## REPORT

# 3D FIB-SEM reconstruction of microtubule–organelle interaction in whole primary mouse β cells

Andreas Müller[1,2,3] , Deborah Schmidt[4,5] , C. Shan Xu[6] , Song Pang[6] , Joyson Verner D'Costa[1,2,3] , Susanne Kretschmar[7], Carla Münster[1,2,3], Thomas Kurth[7] , Florian Jug[4,5,8], Martin Weigert[9]*, Harald F. Hess[6]* , and Michele Solimena[1,2,3,5]*

**Microtubules play a major role in intracellular trafficking of vesicles in endocrine cells. Detailed knowledge of microtubule organization and their relation to other cell constituents is crucial for understanding cell function. However, their role in insulin transport and secretion is under debate. Here, we use FIB-SEM to image islet β cells in their entirety with unprecedented resolution. We reconstruct mitochondria, Golgi apparati, centrioles, insulin secretory granules, and microtubules of seven β cells, and generate a comprehensive spatial map of microtubule–organelle interactions. We find that microtubules form nonradial networks that are predominantly not connected to either centrioles or endomembranes. Microtubule number and length, but not microtubule polymer density, vary with glucose stimulation. Furthermore, insulin secretory granules are enriched near the plasma membrane, where they associate with microtubules. In summary, we provide the first 3D reconstructions of complete microtubule networks in primary mammalian cells together with evidence regarding their importance for insulin secretory granule positioning and thus their supportive role in insulin secretion.**

## Introduction

Cytoskeletal elements, such as microtubules or actin filaments, play a pivotal role in regulating peptide hormone trafficking and secretion in endocrine cells (Rudolf et al., 2001; Park and Loh, 2008; Fourriere et al., 2020). In pancreatic islet β cells, for example, insulin secretion induced upon elevated levels of blood glucose is accompanied by increased polymerization of tubulin (Pipeleers et al., 1976; McDaniel et al., 1980; Heaslip et al., 2014) and a loosening of cortical F-actin (Kalwat and Thurmond, 2013). β cells contain several thousand insulin secretory granules (SGs; Fava et al., 2012) of which 10–20% are dynamically transported along microtubules (Hoboth et al., 2015) and 1–2% are undergoing exocytosis upon glucose stimulation (Rorsman and Renström, 2003). An intact microtubule network (Boyd et al., 1982) as well as motor-mediated transport of SGs are necessary for insulin secretion (Meng et al., 1997; Varadi et al., 2002; Cui et al., 2011), and this active transport increases after stimulation with glucose (Pouli et al., 1998; Hoboth et al., 2015; Müller et al., 2017a). However, in contrast with these data, microtubules have also been postulated to hinder insulin SG transport under low glucose, whereas a loosening of the microtubule network upon glucose stimulation allows SGs to reach the plasma membrane

(Zhu et al., 2015). In view of these considerations, it is therefore crucial to obtain high-resolution data on microtubule remodeling and insulin SG interaction. Previous studies by fluorescence light microscopy showed that the organization of the β cell microtubule network resembles a tangled, nondirectional scaffold (Varadi et al., 2003; Zhu et al., 2015), rather than the radial organization found in tumor cell lines (Meiring et al., 2020). However, even advanced super resolution methods cannot provide the resolution and field of view necessary to accurately reconstruct the 3D microtubule network in whole cells, since microtubules have an outer diameter of ~25 nm and primary β cells are relatively large cells (10–20 μm in diameter) that are densely embedded in the islets of Langerhans. In contrast, EM methods allow nanometer resolution imaging of microtubules together with all other cell structures and organelles. For example, serial-section electron tomography has been used to reconstruct the several thousand microtubules of mitotic spindles in *Caenorhabditis elegans* (Redemann et al., 2017) and the 3D microtubule network of yeast cells in interphase (Höög et al., 2007). Recently, block-face scanning EM has been used to reconstruct microtubules of chemically fixed mitotic spindles of

..........................................................................................................................................................................................................................................................
[1]Molecular Diabetology, University Hospital and Faculty of Medicine, Carl Gustav Carus, Technische Universität Dresden, Dresden, Germany;   [2]Paul Langerhans Institute Dresden of the Helmholtz Center Munich at the University Hospital and Faculty of Medicine, Carl Gustav Carus, Technische Universität Dresden, Dresden, Germany; [3]German Center for Diabetes Research (DZD e.V.), Neuherberg, Germany;   [4]Center for Systems Biology Dresden, Dresden, Germany;   [5]Max Planck Institute of Molecular Cell Biology and Genetics, Dresden, Germany;   [6]Janelia Research Campus, Howard Hughes Medical Institute, Ashburn, VA; [7]Center for Molecular and Cellular Bioengineering, Technology Platform, Technische Universität Dresden, Dresden, Germany;   [8]Fondazione Human Technopole, Milano, Italy;   [9]Institute of Bioengineering, School of Life Sciences, École Polytechnique Fédérale de Lausanne, Lausanne, Switzerland.

*M. Weigert, H.F. Hess, and M. Solimena contributed equally to this paper;   Correspondence to Andreas Müller: andreas.mueller1@tu-dresden.de;   Michele Solimena: michele.solimena@uniklinikum-dresden.de.

HeLa cells followed by spatial analysis (Nixon et al., 2017). However, high-resolution 3D reconstructions of the microtubule network in whole primary mammalian cells in interphase have, to our knowledge, never been reported. Here, we applied high-resolution, near-isotropic focused ion beam scanning electron microscopy (FIB-SEM) to image large volumes of cryo-immobilized, resin-embedded pancreatic islets under different glucose stimuli. We provide the first reconstructions of the full 3D microtubule network within primary mammalian cells, comprising 7 cells in total. We show that β cell microtubules are predominantly not connected to the centrioles and endomembranes and form a nonradial network. We show that glucose stimulation induces microtubule remodeling, which, however, does not change the overall polymerized tubulin density. We additionally reconstruct insulin SGs, Golgi, plasma membranes, nuclei, mitochondria, and centrioles of all cells and perform a comprehensive quantitative spatial analysis of microtubule–organelle interaction. Our data show an enrichment of insulin SGs together with microtubules near the plasma membrane, pointing to the role of microtubules for supporting the secretory functionality in β cells.

## Results and discussion

### High-resolution FIB-SEM resolves organelles and microtubules

To achieve near-isotropic image resolution at high throughput, we used an enhanced FIB-SEM setup (Xu et al., 2017, 2020b) to image large volumes of isolated pancreatic islets of Langerhans treated with low (resting) or high glucose (stimulating insulin secretion) concentrations. Samples were first fixed by high-pressure freezing followed by a novel freeze substitution protocol adapted from Hall et al. (2013) to improve image contrast compared with previously applied freeze substitution protocols (Verkade, 2008; Müller et al., 2017b). Crucially, this improved contrast allowed us to reliably detect microtubules together with all major organelles (Fig. 1 A; and Fig. S1, A, C, and D). We acquired two volumes (one for each glucose condition) of Durcupan-embedded isolated islets containing several full β cells at near-isotropic 4 nm voxel size (Fig. 1 A, Fig. S1 B, and Video 1). Specifically, imaging took approximately 2 wk per stack, resulting in two large volumes with pixel dimensions of 4,900 × 5,000 × 7,570 pixels for the low-glucose and 7,500 × 5,000 × 7,312 pixels for the high-glucose sample with file sizes of 176 and 255 gigabytes, respectively. Acquisition of large volumes combined with high resolution was essential to obtain 3D stacks containing several β cells and to simultaneously resolve microtubules.

We then applied manual as well as machine learning based 3D segmentation to trace microtubules and reconstruct organelles such as the insulin SGs, mitochondria, centrioles with axonemes of the primary cilia, nuclei, the Golgi apparatus, and plasma membrane (Fig. 1 B). Microtubules were segmented manually by creating a skeleton with the KNOSSOS software (Helmstaedter et al., 2011). Plasma membranes, nuclei, and centrioles of individual cells were segmented with Microscopy Image Browser (Belevich et al., 2016). Mitochondria were semi-automatically segmented with ilastik (Berg et al., 2019), whereas insulin SGs were automatically detected with StarDist (Weigert

et al., 2020). The cisternae of the Golgi apparatus were automatically segmented by a U-Net (Ronneberger et al., 2015) that was trained with ground-truth annotations of small crops from the original FIB-SEM volumes. The Golgi apparatus of β cells has in part been reconstructed at high detail (Marsh et al., 2001). Our segmentation was focused on Golgi cisternae and therefore does not include most Golgi vesicles. However, it provides a precise localization of Golgi membrane stacks, which was necessary to investigate microtubule interaction. In total, we obtained full microtubule and organelle segmentations for seven mostly complete β cells (three for the low-glucose and four for the high-glucose condition), which provided the basis for the subsequent visualization and analysis in 3D (Fig. 1 C, Fig. S2, and Video 2). We found that all cells except for one (high-glucose cell 3, Fig. S2) were similar in their general ultrastructure and segmentation features. This particular cell, however, showed signs of stress, such as irregular cell and nucleus shape and a fragmented Golgi apparatus. Nevertheless, we decided to provide all data of this cell and to discuss them in the text where necessary. To facilitate navigation of our results, we developed the tool BetaSeg Viewer, a FIJI plugin based on BigDataViewer (Pietzsch et al., 2015). BetaSeg Viewer enables the joint view of both raw volume and organelle segmentation masks (together with their associated properties), hence an easy and intuitive access to the data (compare Fig. S1, E and F; and Video 3). All segmentation masks and preprocessed data for each individual cell together with the original raw image stacks are publicly available.

### β cell microtubule number and length vary with glucose stimulation

To investigate the remodeling of the microtubule network upon stimulation with glucose, we first quantified basic microtubule properties for low- as well as high-glucose conditions (Fig. 2). The total number of reconstructed microtubules per cell was between 291 and 440 for low-glucose cells, and between 862 and 1,101 for high-glucose cells, implying a substantial increase of microtubule number in glucose-stimulated cells (Table 1). In contrast, the average microtubule length for low-glucose cells ranged from 8.54 to 14.63 µm, which is greater than that of high-glucose cells, where it ranged from 2.66 to 3.25 µm (Fig. 2 B and Table 1). Interestingly, the cumulative microtubule length per cell showed only modest variation between glucose conditions, with 3,018 to 4,260 µm in low-glucose and 2,479 to 3,452 µm in high-glucose cells (Table 1). We additionally computed the tortuosity (curvature) along each microtubule, and found no major differences between microtubules in low- and high-glucose cells (Fig. 2 B). The much shorter microtubule length in high-glucose cells might be due to changes in microtubule dynamics upon glucose stimulation, leading to shorter microtubules (Brouhard and Rice, 2018). High glucose has been proposed to depolymerize microtubules while new microtubules are simultaneously being generated (Zhu et al., 2015), indicating homeostasis of polymerized tubulin. Alternatively, glucose-induced activation of severing enzymes such as spastin and katanin could lead to shorter microtubules. Notably, both spastin and katanin are ATPases Associated with diverse

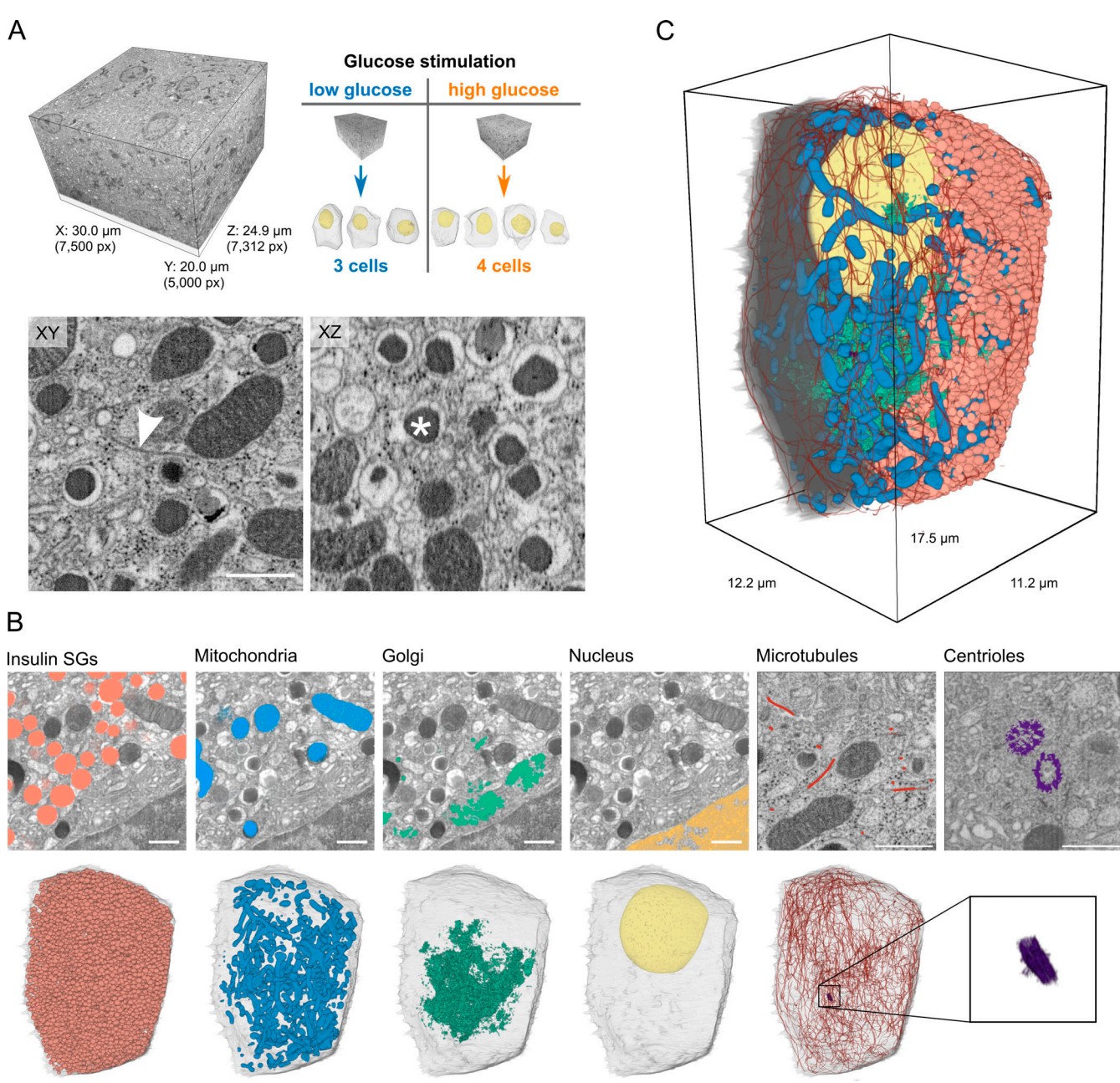

Figure 1. **FIB-SEM volumes of pancreatic β cells and 3D segmentation of microtubules and organelles. (A)** Full FIB-SEM volume of a pancreatic islet (left), one of which was acquired for low- and high-glucose conditions containing three (low) and four (high) complete β cells (right). Shown are the lateral (x-y) and axial (x-z) views of a small crop, highlighting the quasi-isotropic resolution of the FIB-SEM volumes. The arrowhead indicates a microtubule, the asterisk an insulin SG. Scale bar, 500 nm. **(B)** Raw lateral images with segmentation overlay for insulin SGs, mitochondria, Golgi apparatus, nucleus (of the identical region), and microtubules and centrioles (of different regions). Below the overlays, we show 3D renderings of the corresponding organelles of one whole cell (high-glucose condition) accompanied by a transparent rendering of the plasma membrane. Centrioles are magnified in the last panel. Scale bars, 500 nm. **(C)** 3D rendering of one cell containing all segmented organelles. The plasma membrane and insulin SGs were removed in the left half of the cell to help visualize its inner parts.

cellular Activities (AAA ATPases; Hartman et al., 1998; Roll-Mecak and Vale, 2008) that could be sensitive to the increased ATP/ADP ratio induced by glucose stimulation.

## β cell microtubules form nonconnected networks and are enriched near the plasma membrane

We next investigated for each cell the connectivity of the reconstructed microtubule network to different cell organelles.

Our segmentation allowed for precise localization of microtubule ends (Fig. S1 C), enabling us to investigate their connectivity to centrioles and endomembranes. In the case of centrioles, we defined a single microtubule to be centriole-connected or centrosomal if one of its ends is located within the pericentriolar material (∼200 nm around the centrioles). Notably, for all cells, only few microtubules were centriole-connected, with a total number ranging from 8 to 17 in

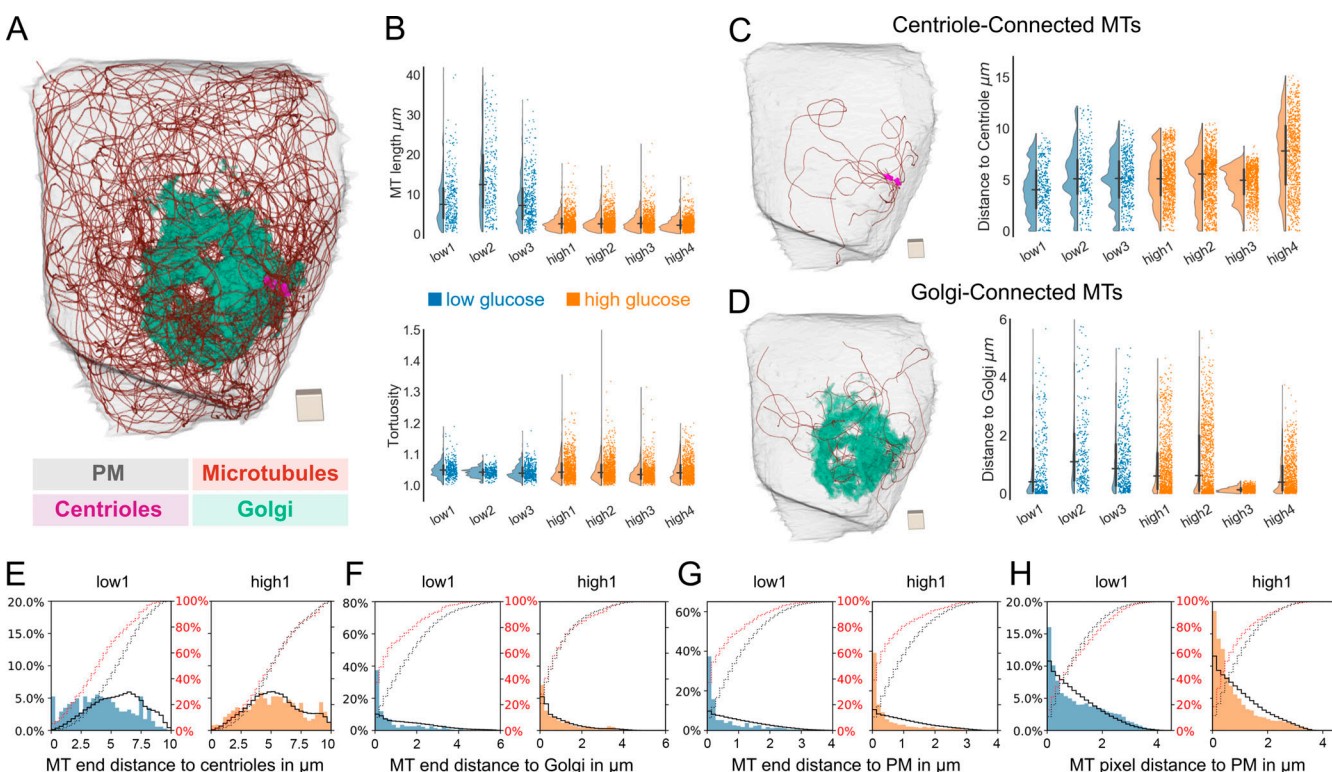

Figure 2. **Microtubule network properties and distance distributions. (A)** Fully reconstructed microtubule network of one β cell with microtubules in red, centrioles in purple, Golgi apparatus in green, and plasma membrane (PM) in gray transparent. Scale, cube with a side length of 1 μm. **(B)** Microtubule (MT) length and tortuosity distribution of all seven analyzed cells (blue, low-glucose cells; orange, high-glucose cells). Horizontal and vertical lines signify the mean and the interquartile range, respectively. **(C)** Rendering of only centriole-connected (centrosomal) microtubules of the same cell as in A. The plots show the distributions of distances of MT ends to centrioles for all analyzed cells. Scale, cube with a side length of 1 μm. **(D)** Rendering of only Golgi-connected microtubules of the same cell as in A. Scale, cube with a side length of 1 μm. The plots show the distributions of distances of the microtubule ends to Golgi for all analyzed cells. **(E)** Distribution of the distance of microtubule ends to centrioles for one representative low-glucose (blue) and one high-glucose (orange) cell with random distributions $\rho_{Cent}$ represented by a black line. Red dotted and black dotted lines represent actual and random cumulative distributions, respectively. **(F)** Distribution of the distance of microtubule ends to Golgi membranes for the same cells as in E with random distributions $\rho_{Golgi}$ represented by a black line. Red dotted and black dotted lines represent actual and random cumulative distributions, respectively. **(G)** Distribution of the distance of microtubule ends to the plasma membrane for the same cells as in E with random distributions $\rho_{PM}$ represented by a black line. Red dotted and black dotted lines represent actual and random cumulative distributions, respectively. **(H)** Distribution of the distance of microtubule pixels to the plasma membrane for the same cells as in E with random distributions $\rho_{PM}$ represented by a black line. Red dotted and black dotted lines represent actual and random cumulative distributions, respectively.

low-glucose and from 6 to 19 in high-glucose cells, corresponding to 2.7 to 3.9% (low glucose) and 0.7 to 1.7% (high glucose) centrosomal microtubules (Fig. 2 C, Table 1, and Video 4). This demonstrates that the microtubule network in mouse β cells is distinctly noncentrosomal irrespective of either glucose condition. Previous studies (Zhu et al., 2015) have indicated that a significant number of microtubules are connected to the Golgi apparatus shortly after microtubule depolymerization by nocodazole. Golgi-derived microtubules originate directly from Golgi membranes (Chabin-Brion et al., 2001; Efimov et al., 2007; Sanders and Kaverina, 2015), and mechanisms for their generation include recruitment of γ-tubulin ring complexes (γ-TuRC) to Golgi membranes with the help of Cytoplasmic Linker Associated Proteins (CLASPs; Efimov et al., 2007) or AKAP450 (Sanders and Kaverina, 2015). Therefore, we chose a distance threshold of 20 nm to define microtubules that are structurally connected to the Golgi apparatus (Golgi connected). We found that between 2 and 10% of all microtubules were Golgi connected (Fig. 2 D, Table 1, and Video 5), while the distance distribution of

microtubule ends to the Golgi peaked at smaller values compared with the centrioles (Fig. 2, C and D; and Fig. S3, A–C). To investigate whether this might be due to the expanded sponge-like geometry of the Golgi rather than the microtubule end distribution itself, we computed for each cell a corresponding random Golgi distribution $\rho_{Golgi}$, i.e., the distribution of distances from the Golgi to all other points inside the cell subvolume excluding the nucleus. That way, $\rho_{Golgi}$ captures the purely geometric effects of the spatial Golgi arrangement and allows an estimate of the distance distribution of the hypothetical case of randomly positioned microtubule ends. If microtubule ends are randomly located in the cell and their positions do not correlate with the location of other organelles, their distribution would roughly match this plot. We found that the distributions of microtubule end distance to the Golgi were slightly shifted toward smaller values compared with $\rho_{Golgi}$ for both glucose conditions (Fig. 2 F; and Fig. S3, B and C). This distribution was heterogeneous between cells. While some cells exhibited a considerable fraction of microtubule ends close to the

Table 1. **Quantitative microtubule and organelle measurements of all cells**

| Glucose status | Low glucose | | | High glucose | | | |
|---|---|---|---|---|---|---|---|
| | Cell 1 | Cell 2 | Cell 3 | Cell 1 | Cell 2 | Cell 3 | Cell 4 |
| Cell volume (µm³) | 799 | 997 | 927 | 1,010 | 834 | 898 | 793 |
| Nucleus volume (µm³) | 115 | 150 | 148 | 126 | 116 | 102 | 106 |
| Nucleus volume (%) | 14.40 | 15.00 | 16.00 | 12.50 | 13.90 | 11.40 | 13.40 |
| Mitochondria total volume (µm³) | 73 | 50 | 104 | 68 | 82 | 44 | 98 |
| Mitochondria total volume (%) | 9.10 | 5.00 | 11.20 | 6.70 | 9.90 | 4.90 | 12.40 |
| Golgi volume (µm³) | 18 | 8 | 8 | 22 | 18 | 12 | 13 |
| Golgi volume (%) | 2.20 | 0.80 | 0.90 | 2.20 | 2.10 | 1.30 | 1.60 |
| Insulin SG number | 5,139 | 9,947 | 4,073 | 9,714 | 7,289 | 13,812 | 9,310 |
| Insulin SG total volume (µm³) | 133 | 210 | 108 | 208 | 153 | 254 | 152 |
| Insulin SG total volume (%) | 16.60 | 21.00 | 11.60 | 20.60 | 18.40 | 28.30 | 19.10 |
| Insulin SG average volume (µm³) | 0.026 | 0.021 | 0.027 | 0.022 | 0.021 | 0.019 | 0.016 |
| Insulin SG average surface area (µm²) | 0.419 | 0.367 | 0.426 | 0.369 | 0.366 | 0.337 | 0.309 |
| Insulin SG average diameter (µm) | 0.409 | 0.383 | 0.413 | 0.384 | 0.384 | 0.368 | 0.355 |
| Insulin SG assoc. with MT in % of all SG | 30.70 | 34.40 | 37.30 | 29.10 | 39.00 | 28.70 | 27.00 |
| Assoc. insulin SG close to PM (≤500 nm) in % | 53.00 | 45.30 | 59.20 | 56.60 | 55.40 | 51.20 | 62.10 |
| Not assoc. insulin SG close to PM (≤500 nm) in % | 50.60 | 44.40 | 51.50 | 41.70 | 49.50 | 25.30 | 59.60 |
| Microtubule number | 342 | 291 | 440 | 1,007 | 1,101 | 862 | 932 |
| Microtubule average length (µm) | 8.82 | 14.63 | 8.54 | 3.1 | 3.14 | 3.25 | 2.66 |
| Microtubule cumulative length (µm) | 3,018 | 4,260 | 3,756 | 3,115 | 3,452 | 2,799 | 2,479 |
| Microtubule average tortuosity | 1.05 | 1.04 | 1.04 | 1.05 | 1.05 | 1.04 | 1.05 |
| Microtubule centriole-connected % | 3.50 | 2.70 | 3.90 | 0.70 | 1.70 | 0.70 | 1.30 |
| Microtubule Golgi-connected % | 9.60 | 0.30 | 0.50 | 3.70 | 10.20 | 3.00 | 3.20 |

Volumes of the analyzed cells, nuclei, mitochondria, Golgi apparati, insulin SGs. Insulin SG numbers and individual volumes. Numbers and percentages of microtubule-associated and –not associated SGs. Numbers, average and cumulative lengths, and tortuosity of microtubules. Percentages of centrosomal and Golgi-connected microtubules. assoc., associated; MT, microtubule; PM, plasma membrane.

Golgi (low glucose 1, high glucose 2 and 4), such an enrichment was not observed in other cells (low glucose 2 and 3, high glucose 1; see Fig. S3 B). A high density of microtubule ends that are close, yet structurally not connected, to Golgi membranes indicates that these microtubules might have originated from the Golgi, but have subsequently lost their connection to it. In contrast, the distance distribution of microtubule ends to the centrioles was more similar to the respective $\rho_{Cent}$ (Fig. 2 E and Fig. S3 A). One cell (high glucose cell 3) showed a highly fragmented Golgi (Fig. S2), which led to a strongly enriched distribution of microtubule ends close to Golgi membranes (Fig. 2 D; Fig. S2; and Fig. S3, B and C). However, the fraction of Golgi-connected microtubules was similar to that of the other cells, and the distribution of microtubule ends can be explained by the fragmentation of the Golgi apparatus itself. Surprisingly, independent of glucose concentration, there was stronger enrichment of microtubule ends close the plasma membrane compared with its random distribution $\rho_{PM}$ in all cells (Fig. 2 G and Fig. S3 D). Moreover, microtubule pixels were also enriched near the plasma membrane (Fig. 2 H and Fig. S3 E). Notably we did not find major differences in polymerized tubulin density between

glucose conditions (Fig. 2 H and Fig. S3 E). Our findings show that although a small number of microtubules is connected to centrioles and Golgi apparatus, over 80% of all microtubules are disconnected from these compartments. Although both Golgi apparatus and centrioles could still serve as microtubule organizing centers, our data suggest either other places of microtubule generation, or severing of microtubules after their generation. Notably, a significant body of literature indicates that microtubules can appear spontaneously in the cytosol (Vorobjev et al., 1997; Yvon and Wadsworth, 1997). To stabilize these microtubules, proteins such as CAMSAP2 (Jiang et al., 2014) or anchoring proteins such as Ninein (Mogensen et al., 2000) are necessary and might play a role in β cells. Finally, the observed remodeling toward shorter microtubules under glucose stimulation did not alter the polymerized tubulin density in general.

## Organelle fractions are heterogeneous across cells
To gain insight into the role of microtubules in organelle positioning, we reconstructed insulin SGs, mitochondria, plasma membranes, centrioles with axonemes of the primary cilia, Golgi

apparati, and nuclei of all seven cells (Fig. S2, Table 1, and Video 2). Quantification of these organelles showed a clear heterogeneity among the cells, independent of glucose stimulation (Table 1). The volumes of the cells were variable, ranging from 799 to 997 µm³ in the low-glucose cells and from 793 to 1,010 µm³ in the high-glucose cells. Mitochondria volumes ranged from 50 to 104 µm³ for low-glucose and from 44 to 98 µm³ for high-glucose cells. The volume of the nuclei ranged from 102 to 150 µm³ in the six cells containing complete nuclei. The number of insulin SGs was especially variable, with only 4,073 SGs in one of the low-glucose cells and up to 13,812 SGs in one high-glucose cell. These data span the entire range of values that so far had been calculated for an average β cell using either stereological methods (Dean, 1973; Olofsson et al., 2002; Shomorony et al., 2015) or in silico modeling (Fava et al., 2012). Our data clearly demonstrate a strong heterogeneity of β cells in the number of their SGs independent of glucose stimulation. This is in agreement with Noske et al. (2008), who observed substantial organelle heterogeneity between two β cells reconstructed with serial section electron tomography (however, without considering microtubules due to limited resolution). To investigate if the differences in SG number were correlated with cell size, we calculated the volume fractions of insulin SGs, and additionally of the nuclei, Golgi apparati, and mitochondria. While nuclei and Golgi fractions were relatively stable across cells, volume fractions of mitochondria and insulin SGs were highly variable even within the same glucose conditions (compare Fig. 3 A): for insulin SGs, the volume fraction ranged from 11.6 to 28.8%, demonstrating that insulin SG number does not directly correlate with cell volume.

### Insulin SGs are enriched near the plasma membrane, but not the nucleus, independent of glucose concentration

Next, we wondered if the observed β cell heterogeneity would also be reflected in insulin SG distribution and interaction with other organelles. Our 3D high-resolution reconstructions allowed for a quantitative analysis of insulin SGs and their spatial distribution and interaction with the plasma membrane, nucleus, and Golgi apparatus (Fig. 3 B). In total, we analyzed the precise segmentation masks of >60,000 insulin SGs for all cells. First, we calculated SG volumes, surface areas, diameters, and sphericity for all SGs in our cells (Fig. 3 C; and Fig. S3, F and G). The average SG volume ranged between 0.016 to 0.026 µm³ and the SG surface area between 0.309 to 0.426 µm². Insulin SGs in all cells were close to spherical in shape (Fig. S3 G) with a mean diameter between 355 and 413 nm (Fig. 3 C and Table 1), comparable to values obtained by stereological methods (Dean, 1973; Olofsson et al., 2002). Next, we investigated the distance distribution of insulin SGs to the plasma membrane. As before, we also computed the corresponding random distributions $\rho_{PM}$ for comparison. As can be inferred from Fig. 3 D and Fig. S3 H, especially SGs closest to the plasma membrane are substantially over-represented when compared with $\rho_{PM}$. We observed this phenomenon in all cells, including the ones with a smaller fraction of insulin SGs. In contrast, SG distances from the nucleus followed approximately the random distributions $\rho_{Nuc}$ in all cells (Fig. 3 E and Fig. S3 I). Interestingly, the insulin SG distance to Golgi distribution in low-glucose cells followed the

random distributions $\rho_{Golgi}$, whereas under high-glucose conditions, SGs were slightly enriched near the Golgi, indicative of the production of new SGs (Fig. 3 F and Fig. S3 J). We therefore conclude that SGs are enriched near the plasma membrane under both low- and high-glucose conditions, but not near the nucleus.

### Insulin SGs associate with microtubules close to the plasma membrane

The strong enrichment of insulin SGs as well as microtubules at the plasma membrane led us to investigate the distance distribution and direct association of both organelles (Fig. 4 A). Interestingly, insulin SGs were strongly enriched close to microtubules when compared with random microtubule distributions $\rho_{MT}$ in all cells independent of glucose concentration (Fig. 4 B and Fig. S3 K). Furthermore, a high percentage of SGs was directly associated with (<20 nm) microtubules in both glucose conditions. We found between 30.7 and 37.3% of associated SGs in low-glucose cells and between 27.0 and 39.0% in high glucose cells (Fig. 4 C, Table 1, Fig. S2, and Video 6). This corresponded to 1.95–4.04 microtubule-associated SGs per cubic micrometer of cytoplasmic volume in low-glucose (rested) cells and 3.20–4.98 microtubule-associated SGs per cubic micrometer of cytoplasmic volume in glucose-stimulated cells. Furthermore, the majority of microtubule-associated SGs were located close to the plasma membrane (Fig. 4 D). Interestingly, the fraction of microtubule-associated SGs at the plasma membrane (within a distance of 500 nm) was as high or higher than the fraction of not associated SGs, independent of glucose stimulation (Table 1). The distance distribution of microtubule-associated and –not associated SGs from the nucleus and Golgi apparatus was more heterogeneous, with some cells showing a higher percentage of associated SGs close to these organelles and others a higher percentage of not connected SGs (Fig. 4, E and F). Together with the data on microtubule and SG distribution, these findings indicate an active role of microtubules in maintaining the enrichment of insulin SGs in the cell periphery.

### Summary

We provide here the first 3D reconstructions of all microtubules in primary mammalian cells in interphase using pancreatic β cells as a model system. Although EM can only give snapshots of the cell's ultrastructure, our data provide unprecedented insights into the 3D microtubule organization and organelle interaction with nanometer precision (Video 7). We found that a minority of microtubules are connected to the centrioles and Golgi apparatus, while most of the microtubules appear to be freely positioned within the cytosol. The presence of largely nonradial, disconnected microtubule networks is a key feature of differentiated cells (Muroyama and Lechler, 2017) and implies a major microtubule remodeling during β cell development. Notably, immature β cells that proliferate have a lower secretory capacity compared with mature β cells (Salinno et al., 2019). Microtubule remodeling combined with a change in microtubule purpose, i.e., from spindle formation to membrane trafficking, could contribute to the increased β cell functionality. We furthermore found substantial differences in microtubule number and length between low- and high-glucose conditions,

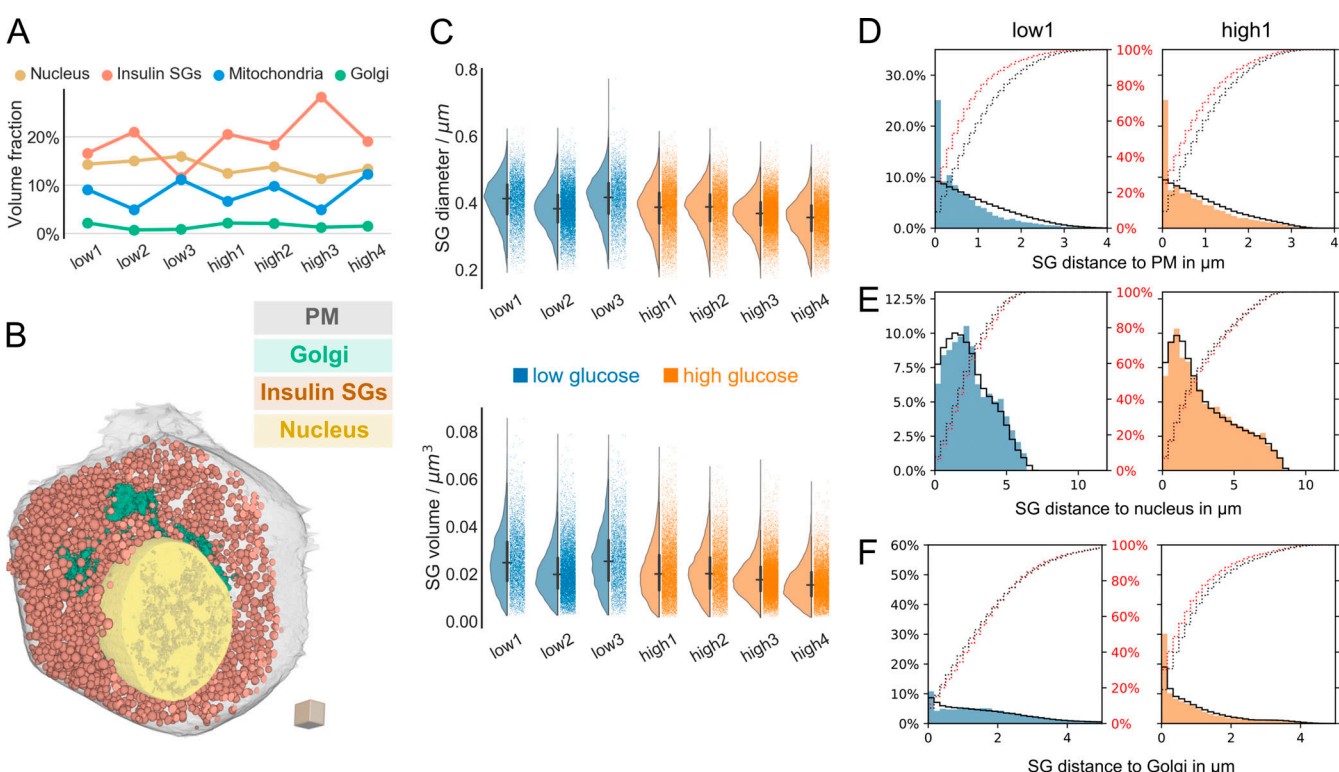

Figure 3. **Insulin SG properties and distance distributions. (A)** Volume fraction (percentage) of segmented organelles for all analyzed cells. **(B)** 3D rendering of one β cell with plasma membrane (transparent gray), insulin SGs (orange), Golgi apparatus (green), and nucleus (yellow). Scale: cube with a side length of 1 μm. **(C)** Diameter and volume distributions of insulin SGs of all seven analyzed cells (blue, low-glucose cells; orange, high-glucose cells). Horizontal and vertical lines signify the mean and the interquartile range, respectively. **(D)** Distribution of the distance of insulin SGs to the plasma membrane (PM) for one representative low-glucose (blue) and one high-glucose (orange) cell with random distributions $\rho_{PM}$ represented by a black line. Red dotted and black dotted lines represent actual and random cumulative distributions, respectively. **(E)** Distribution of the distance of insulin SGs to the nucleus for one representative low-glucose (blue) and one high-glucose (orange) cell with random distributions $\rho_{Nuc}$ represented by a black line. Red dotted and black dotted lines represent actual and random cumulative distributions, respectively. **(F)** Distribution of the distance of insulin SGs to the Golgi apparatus for one representative low-glucose (blue) and one high-glucose (orange) cell with random distributions $\rho_{Golgi}$ represented by a black line. Red dotted and black dotted lines represent actual and random cumulative distributions, respectively.

suggesting a major remodeling of the microtubule network upon glucose stimulation. However, in contrast to Zhu et al. (2015), we did not measure changes in density of polymerized tubulin between glucose conditions, questioning the proposed hindering role of microtubules for SG mobility in low glucose. Moreover, in all cells, microtubules as well as insulin SGs were enriched near the plasma membrane, implying the role of microtubules in maintaining organelle positioning (de Forges et al., 2012). Interestingly, the fraction of insulin SGs directly connected to microtubules was between 20 and 40% regardless of glucose treatment. Thus, microtubules likely maintain a high density of insulin SGs near the plasma membrane under low- as well as under high-glucose conditions, keeping insulin SGs in close proximity to secretion sites. In conclusion, our data provide quantitative insights into the ultrastructure of β cells and the reciprocal interaction among their compartments at an unprecedented resolution. This detailed knowledge will facilitate our understanding of β cell function and insulin trafficking.

### Data and software availability
Segmentation masks and crops of analyzed β cells have been binned to near 16 nm isotropic voxel size and are available

for download via https://betaseg.github.io/. The FIJI plugin Beta-Seg Viewer and the plugin for importing KNOSSOS skeletons can be downloaded via the update site https://sites.imagej.net/betaseg/.

## Materials and methods

### Islet isolation and culture
Pancreatic islets of 9-wk-old C57BL/6 mice were isolated as previously described (Gotoh et al., 1985). They were cultured overnight in standard culture media (Roswell Park Memorial Institute 1640 [Gibco] with 10% FBS, 20 mM Hepes, and 100 U/ml each of penicillin and streptomycin) containing 5.5 mM glucose. Prior to high-pressure freezing, the islets were subjected to 1 h incubation in Krebs–Ringer buffer containing either 3.3 mM or 16.7 mM glucose. All animal experiments were performed according to guidelines of the Federation of European Laboratory Animal Science Associations (FELASA) and recommendations and are covered by a respective licenses for those experiments from the local authorities. Facilities for animal keeping and husbandry are certified and available with direct access on campus in Dresden (including facilities at Paul

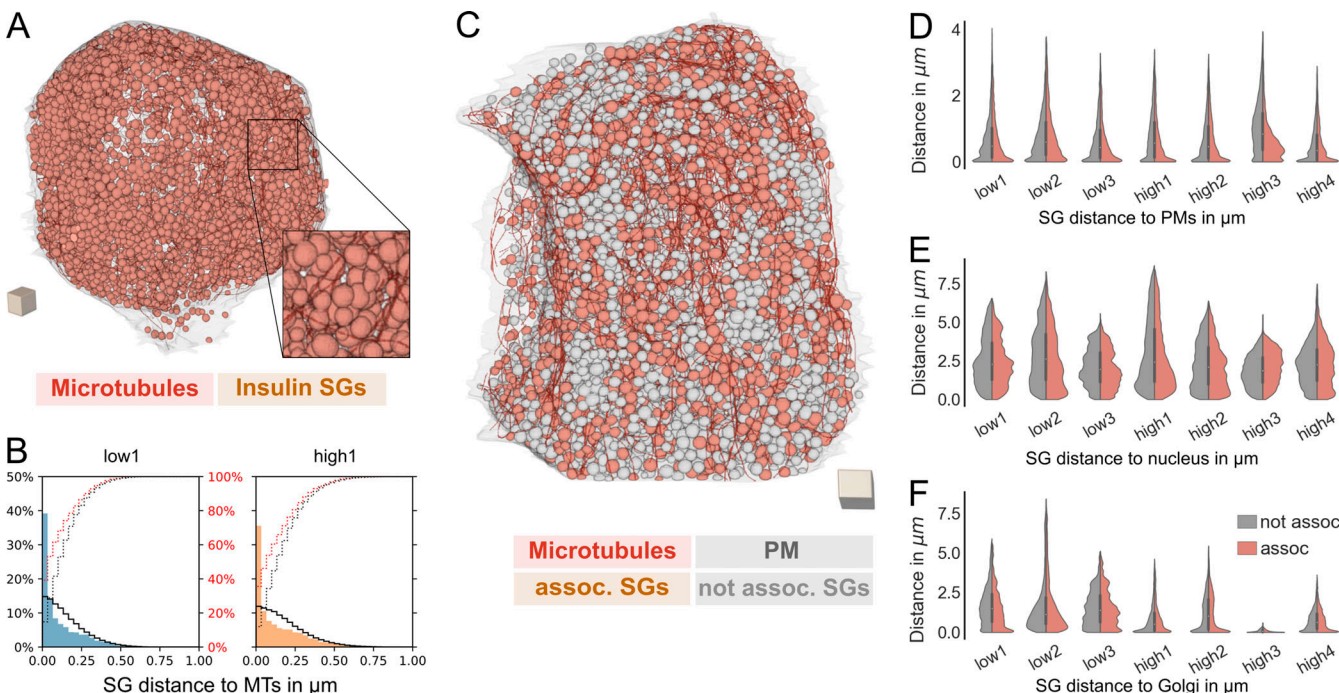

Figure 4. **Spatial association between microtubules and insulin SGs. (A)** 3D rendering of a cell with plasma membrane (transparent gray), microtubules (red), and insulin SGs (orange). Inset shows a magnified region. Scale: cube with a side length of 1 μm. **(B)** Distribution of the distance of insulin SGs to microtubules (MTs) for one representative low-glucose (blue) and one high-glucose (orange) cell with random distributions $\rho_{MT}$ represented by a black line. Red dotted and black dotted lines represent actual and random cumulative distributions, respectively. **(C)** 3D rendering of a cell with plasma membrane, microtubule-associated SGs (orange), not associated SGs (light gray), and microtubules (red). Scale: cube with a side length of 1 μm. **(D)** Violin plots depicting the distance of associated and not associated insulin SGs to the plasma membrane (PM). **(E)** Violin plots depicting the distance of associated and not associated insulin SGs to the nucleus. **(F)** Violin plots depicting the distance of associated and not associated insulin SGs to the Golgi apparatus. assoc., associated.

Langerhans Insitute Dresden and Max Planck Insitute of Cell Biology and Genetics). Licenses for animal experiments are approved by the State Directorate Saxony under license number DD24.1-5131/450/6.

**High-pressure freezing, freeze substitution, and embedding**
Islets were frozen with a Leica EM ICE high-pressure freezer (Leica Microsystems) and kept in liquid nitrogen until freeze substitution. They were substituted as previously published (Müller et al., 2017b) or according to a novel protocol: first, the samples were substituted in a cocktail containing 2% osmium-tetroxide, 1% uranylacetate, 0.5% glutaraldehyde, 5% $H_2O$ (according to Buser and Walther, 2008) in acetone with 1% methanol at –90°C for 24 h. The temperature was raised to 0°C over 15 h followed by four washes with 100% acetone for 15 min each and an increase in temperature to 22°C. Afterwards, the samples were incubated in 0.2% thiocarbohydrazide in 80% methanol at RT for 60 min followed by 6 × 10 min washes with 100% acetone. The specimens were stained with 2% osmium-tetroxide in acetone at RT for 60 min followed by incubation in 1% uranylacetate in acetone plus 10% methanol in the dark at RT for 60 min. After four washes in acetone for 15 min each, they were infiltrated with increasing concentrations of Durcupan resin in acetone followed by incubation in pure Durcupan and polymerization at 60°C. For quality control, the blocks were sectioned with a Leica LC6 ultramicrotome (Leica Microsystems), and 300-nm sections were put on slot grids containing

a Formvar film. Tilt series ranging from –63° to +63° were acquired with a F30 EM (Thermo Fisher Scientific) and reconstructed with the IMOD software package (Kremer et al., 1996).

**FIB-SEM imaging**
Prior to FIB milling, small vertical posts were trimmed to the region of interest guided by x-ray tomography data obtained by a Zeiss Versa XRM-510 and optical inspection under a microtome. A thin layer of conductive material of 10 nm gold followed by 100 nm carbon was coated on the trimmed samples using a Gatan 682 High-Resolution Ion Beam Coater. The coating parameters were 6 keV, 200 nA on both argon gas plasma sources, 10 rpm sample rotation with 45° tilt. The coated samples were imaged using a customized FIB-SEM with a Zeiss Capella FIB column mounted at 90° onto a Zeiss Merlin SEM. Details of the enhanced FIB-SEM systems were previously described (Xu et al., 2017, 2020a,c). Each block face was imaged by a 140 pA electron beam with 0.9 keV landing energy at 200 kHz. The x-y pixel size was set at 4 nm. A subsequently applied focused Ga+ beam of 15 nA at 30 keV strafed across the top surface and ablated away 4 nm of the surface. The newly exposed surface was then imaged again. The ablation–imaging cycle continued about once every 3–4 min for 2 wk to complete FIB-SEM imaging one sample. The sequence of acquired images formed a raw imaged volume, followed by post-processing of image registration and alignment using a Scale Invariant Feature Transform–based algorithm. The

actual z-step was estimated by the changes of SEM working distance and FIB milling position. Specifically, the voxel dimensions were 4 × 4 × 4.24 nm (low-glucose volume) and 4 × 4 × 3.40 nm (high-glucose volume). The image stacks were rescaled to form 4 × 4 × 4 nm isotropic voxels, which can be viewed in any arbitrary orientations.

### Segmentation of FIB-SEM data

Due to the differences in organelle morphology and involved length scales, segmentation of the different organelles poses specific challenges that we approached with a dedicated method for each organelle class: large and contrast-rich organelles such as plasma membrane, nucleus, and mitochondria allowed fast semi-manual tracing or admitted relatively uncomplicated pixel-wise classification methods. The crowded distribution of insulin SGs and the convoluted structure of the Golgi complex required a two-stage process to first generate and curate training data for a subsequent final machine learning–based segmentation step (see Golgi apparatus and SGs paragraphs below). Microtubules posed the biggest segmentation challenge due to their small diameter (25 nm) and relatively low electron density. We thus opted in this case to perform laborious yet accurate manual tracing.

#### Microtubules

Microtubules were traced manually by creating a skeleton with the KNOSSOS software (Helmstaedter et al., 2011). The original 4 nm isotropic voxel size of the dataset was used. Tracing was performed by two experts. Final corrections were done by one of them. The final skeletons were scaled to reach a final microtubule thickness of 25 nm. The manual tracing and curating of the microtubule skeleton required ~50 h per cell.

#### Plasma membrane, nucleus, and centrioles

Plasma membrane, nucleus, and centrioles were segmented manually with the Microscopy Image Browser (Belevich et al., 2016) using interpolation every 10 slices. Centrioles were roughly segmented followed by thresholding. The same was done for the nucleus.

#### Golgi apparatus

We used ilastik to create a preliminary foreground mask of 10 crops (128 × 128 × 128 pixels each) from different cells. After manual curation, these labeled crops were used to train a three-class U-Net (Ronneberger et al., 2015), which was then applied to all full volumes. We used data augmentation (flips, elastic deformations, Gaussian noise) during training. The volumes were binned by four, resulting in 16-nm isotropic voxel size datasets.

#### Mitochondria

Mitochondria were segmented with ilastik (Berg et al., 2019), which employs a random forest classifier that predicts for every pixel the probability to belong either to the background or to the mitochondria class. Additionally, we used the autocontext function of ilastik, which resulted in improved segmentation masks. The volumes were binned fourfold, resulting in 16-nm isotropic voxel size datasets. Masks were then exported as hdf5 files.

#### SGs

Deep learning has been successfully applied to detect insulin SGs in 2D images (Zhang et al., 2019). Due to the packed and dense distribution of SGs, accurate instance segmentation (i.e., assigning each SG a unique label) is very challenging. To address this, we first created preliminary instance masks with ilastik and a subsequent watershed transform and used them as preliminary ground-truth for training a 3D StarDist model (Weigert et al., 2020). After one round of training with these preliminary (and imperfect) masks, we manually curated five small crops of the first StarDist result (172 × 172 × 172 pixels each) that served as precise ground-truth for the second and final round of StarDist training. We used data augmentation (flips, elastic deformations, Gaussian noise, and intensity shift) during training. The volumes were again binned fourfold, resulting in 16-nm isotropic voxel size datasets.

### Data analysis and visualization

#### Segmentation postprocessing

Segmentation masks were first exported as hdf5 or Amira files, then imported into Fiji, and finally saved as TIFF files (Schindelin et al., 2012). Microtubule skeletons generated in KNOSSOS were imported into Fiji using a custom plugin and also saved as TIFF files.

#### Distance and connectivity calculations

Based on the segmentation and labeling data, we calculated distance maps of organelle pixels and distance and connectivity relations between individual labels with ImageJ2 (Rueden et al., 2017) and ImgLib2 (Pietzsch et al., 2012). Microtubules were categorized as centrosomal if the distance of one of their ends to the centrioles was <200 nm. We considered a SG to be associated with a microtubule if the minimal distance between SG pixels and a microtubule was <20 nm. Microtubule ends were considered as connected to the Golgi if their distance was <20 nm.

#### Distribution plots

We used Python-based tools (Harris et al., 2020; Hunter, 2007; Virtanen et al., 2020; W. McKinney. 2010. Proceedings of the 9th Python in Science Conference.) to process and visualize all distributions. When plotting microtubule end distances, e.g., to the membrane, from the two ends of each microtubule, only the one closer to the membrane was taken into account. To set the distributions of distance measurements into perspective, we additionally plotted the nonzero distance distribution of the respective organelle in the cell, excluding the volume occupied by the nucleus. For instance, when plotting the distance of microtubule ends to the Golgi, the second histogram line in black accounts for the distances of all pixels in the cell to the Golgi, excluding any pixels belonging to the nucleus or to the Golgi itself. We refer to these plots as random distributions $\rho_X$ where X signifies the organelle with respect to which the distances were calculated (e.g., $\rho_{Golgi}$, $\rho_{MT}$, etc.).

#### Fiji plugins

The plugins available in Fiji via our update site, https://sites. imagej.net/betaseg, BetaSeg Viewer, and the plugin for importing

KNOSSOS skeletons are both based on the SciJava plugin framework (https://scijava.org/). The viewer utilizes a preliminary version of LabelEditor (https://github.com/juglab/LabelEditor), a novel layer on top of BigDataViewer (Pietzsch et al., 2015) for displaying and interacting with labeling and label attributes. These attributes, e.g., microtubule length or distances between organelles, can directly be plotted with JFreeChart (http://www.jfree.org/jfreechart/).

### BetaSeg Viewer instructions
Fiji/ImageJ users can install the BetaSeg Viewer by adding the update site https://sites.imagej.net/BetaSeg via the Help>-Update... in the menu. The data folder of at least one cell needs to be downloaded from https://betaseg.github.io/. Afterwards, one can start the viewer from the menu by clicking Analysis>BetaSeg and then pointing to the directory of one cell in the following popup. A new window with a BigDataViewer area on the left and a list of data components on the right will open, and the EM source of the cell will be automatically displayed. The list of data components allows one to interactively show or hide segmentations as well as analysis results, e.g., the length or tortuosity of microtubules. These properties are displayed via lookup tables in the viewer on top of the EM source. Colors can be adjusted. Individual values of, e.g., single microtubules can be displayed by clicking the respective object. The analysis data can also be displayed as a histogram or table, available via the (...) button on the right of the specific data item.

### Visualization and 3D rendering
Overlays of raw data and segmentations were visualized with 3Dscript (Schmid et al., 2019). 3D rendering was done with ORS Dragonfly (https://theobjects.com/dragonfly) or blender (https://www.blender.org).

### Online supplemental material
Fig. S1 shows raw FIB-SEM data and workflow for sample preparation, imaging, segmentation, and data integration with BetaSeg Viewer. Fig. S2 shows 3D renderings of all cells and organelles/organelle subtypes analyzed in this study. Fig. S3 shows microtubule and insulin SG analysis for all cells. Video 1 shows raw FIB volume. Video 2 shows β cell microtubule and organelle segmentation. Video 3 shows the BetaSeg Viewer. Video 4 shows the microtubule network and centrosomal microtubules. Video 5 shows the microtubule network and Golgi-connected microtubules. Video 6 shows microtubule-associated and –not associated insulin SGs. Video 7 shows 3D animation of all segmented β cells.

## Acknowledgments

We thank the EM facility of Max Planck Institute of Cell Biology and Genetics for their services. We thank Katja Pfriem for administrative assistance. We thank members of the Paul Langerhans Institute Dresden for valuable feedback. We thank Dr. Ricardo Henriques for providing the BioRxiv LaTeX template. We are grateful to Dr. Jaber Dehghany (formerly Helmholtz Zentrum für Infektionsforschung, Braunschweig), Dr. Erin Tranfield (Instituto Gulbenkian de Ciência, Portugal), Dr. Aubrey Weigel, Dr. Jan Funke, Dr. Stephan Saalfeld (all Howard Hughes Medical Institute Janelia), and Dr. Carl Modes (Center for Systems Biology Dresden) for helpful discussions. We thank Prof. Thomas Müller-Reichert and members of his laboratory (Dr. Gunar Fabig, Robert Kiewiz, Dr. Peter Horvath) and Dr. Robert Haase (Max Planck Institute of Cell Biology and Genetics) for constructive feedback. We thank Dr. Uwe Schmidt (Center for Systems Biology Dresden) for advice on ilastik.

This work was supported with funds to M. Solimena from the German Center for Diabetes Research (DZD e.V.) by the German Ministry for Education and Research (BMBF), from the German-Israeli Foundation for Scientific Research and Development (GIF; grant I-1429-201.2/2017), from the German Research Foundation (DFG) jointly with the Agence Nationale de la Recherche (grant SO 818/6-1), and from the Innovative Medicines Initiative 2 Joint Undertaking under grant agreements 115881 (RHAPSODY) and 115797 (INNODIA), which include financial contributions from the European Union's Framework Program Horizon 2020, EFPIA, and the Swiss State Secretariat for Education, Research and Innovation under contract 16.0097, as well as JDRF International and the Leona M. and Harry B. Helmsley Charitable Trust. A. Müller was the recipient of a MeDDrive grant (60417) from the Carl Gustav Carus Faculty of Medicine at TU Dresden. D. Schmidt and F. Jug were supported by the German Research Foundation (DFG; grant JU3110/1-1). T. Kurth and the EM facility of the Center for Molecular and Cellular Bioengineering are supported by the European Fund for Regional Development. M. Weigert was supported by a generous donor represented by CARIGEST SA. C.S. Xu, S. Pang, and H.F. Hess are supported by the Howard Hughes Medical Institute.

The authors declare no competing financial interests.

Author contributions: conceptualization, A. Müller and M. Solimena; investigation, A. Müller, D. Schmidt, S. Pang, C.S. Xu, M. Weigert, J.V. D'Costa, S. Kretschmar, C. Münster, T. Kurth, F. Jug, H.F. Hess, and M. Solimena; formal analysis, A. Müller, D. Schmidt, M. Weigert, and J.V. D'Costa; software, D. Schmidt; visualization, D. Schmidt, A. Müller, and M. Weigert; writing—original draft, A. Müller, D. Schmidt, M. Weigert, and M. Solimena; writing—review and editing, A. Müller, D. Schmidt, C.S. Xu, M. Weigert, J.V. D'Costa, T. Kurth, F. Jug, H.F. Hess, and M. Solimena; and funding acquisition, A. Müller and M. Solimena.

Submitted: 7 October 2020

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

# Supplemental material

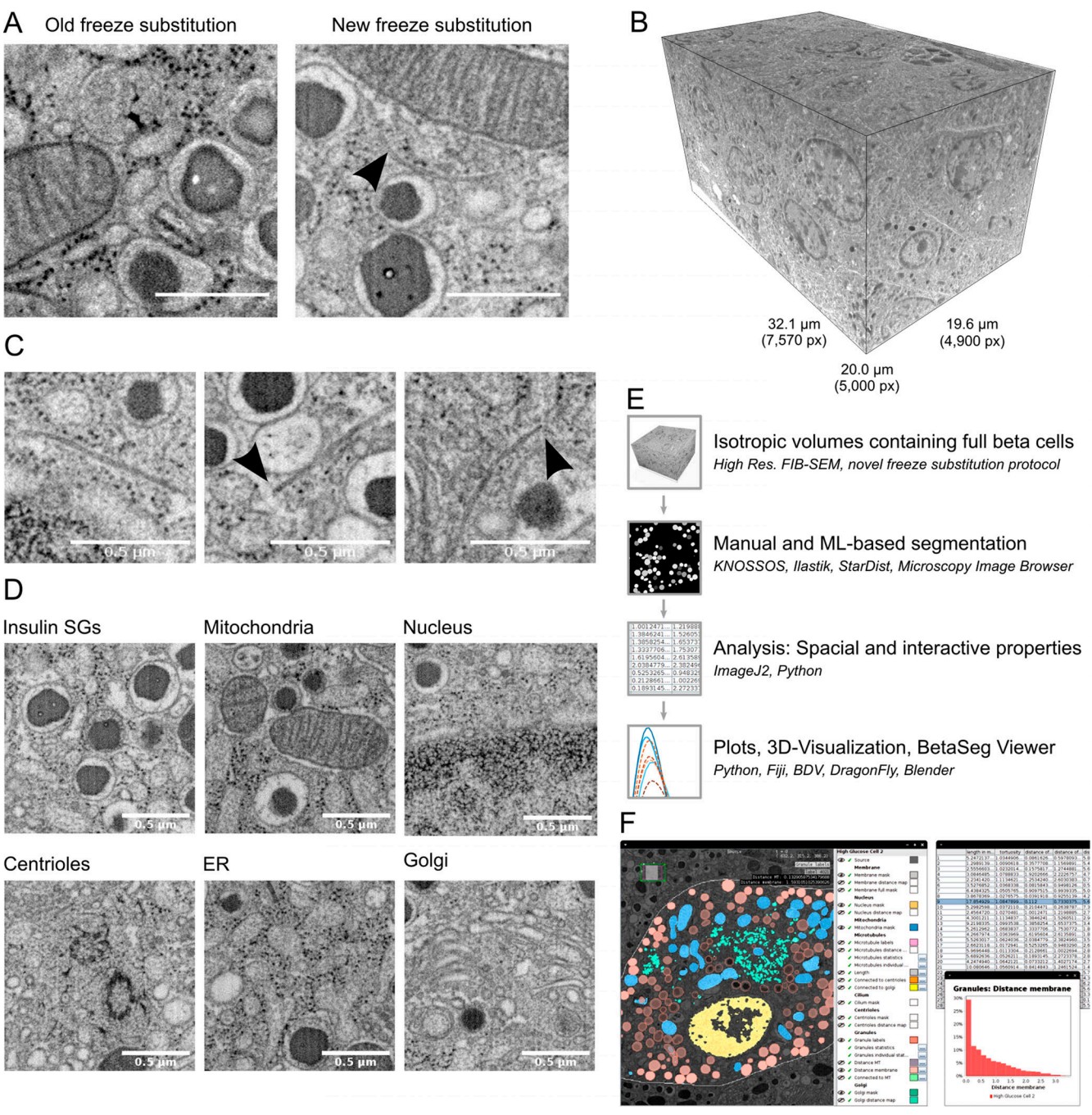

Figure S1. **Raw FIB-SEM data and workflow for sample preparation, imaging, segmentation, and data integration within BetaSeg Viewer.** **(A)** Snapshots of samples prepared according to the old and new freeze substitution protocol. Arrowhead, microtubule. Scale bar, 500 nm. **(B)** Full raw volume of the low-glucose dataset with pixel and micrometer dimensions. **(C)** Detailed views of microtubules with arrowheads pointing to microtubule ends. **(D)** Snapshots of ultrastructural details: insulin SGs, mitochondria, nucleus, centrioles, ER, and Golgi apparatus. Scale bar, 500 nm. **(E)** Workflow for sample preparation, imaging, segmentation, and data integration within BetaSeg Viewer: isotropic volumes of cryo-immobilized, freeze-substituted, and resin-embedded pancreatic islets were acquired with FIB-SEM followed by manual and machine learning segmentation, 3D data analysis, integration into BetaSeg Viewer, and 3D visualization. **(F)** Screenshot of BetaSeg Viewer with a slice through an overlay of the raw volume of one β cell with the corresponding segmentation masks, a table depicting quantitative data, and a plot showing the distance of insulin SGs to the plasma membrane generated with BetaSeg Viewer. ML, machine learning; Res., resolution.

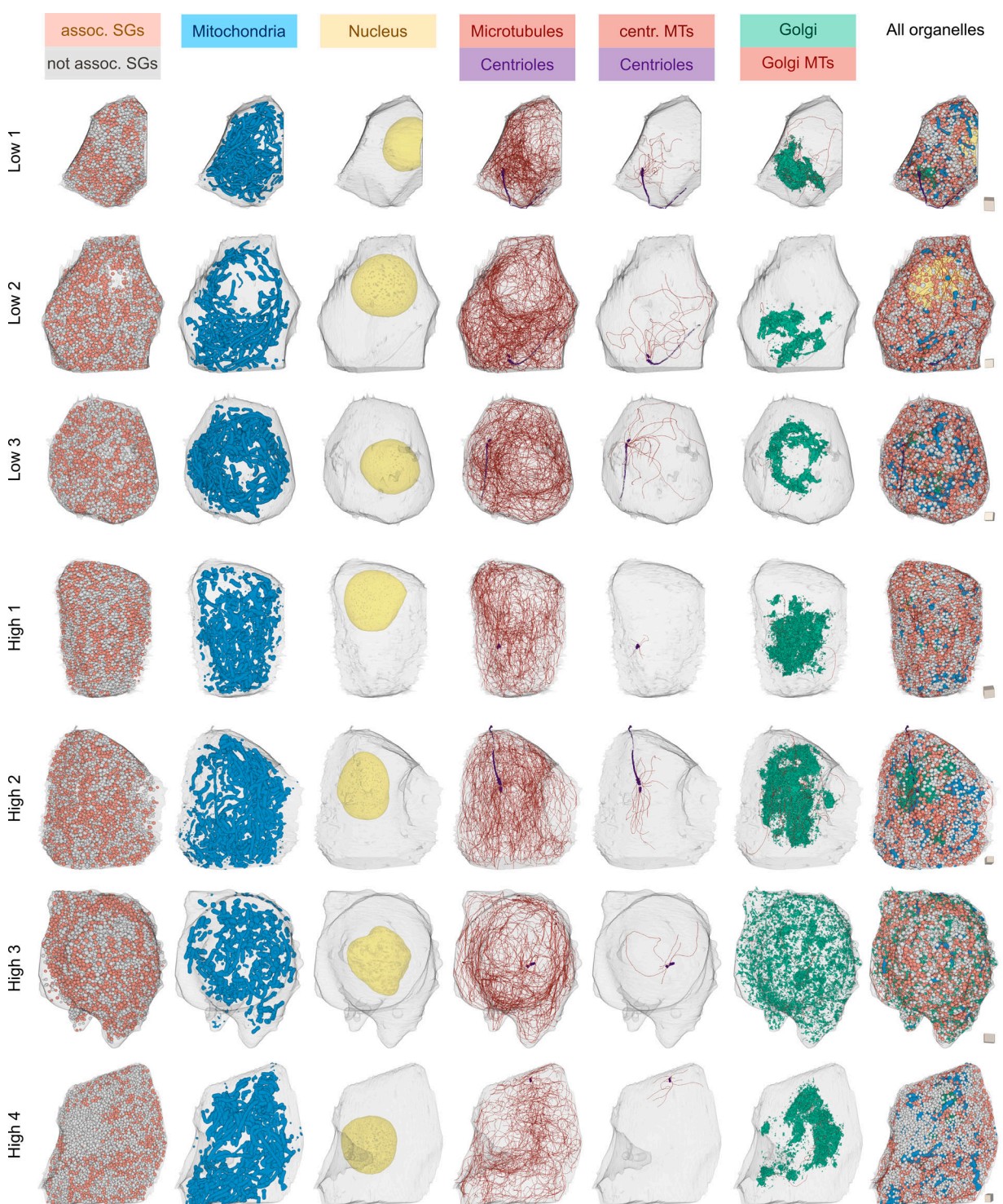

Figure S2. **3D renderings of all β cells and organelles/organelle subtypes analyzed in this study.** Color-coded are microtubule-associated and –not associated SGs, mitochondria, nuclei, microtubules, centrioles, centrosomal microtubules, Golgi apparati, and Golgi microtubules. Cubes on the right of each cell have a side length of 1 μm for scaling. assoc., associated; MT, microtubule; centr., centrosomal.

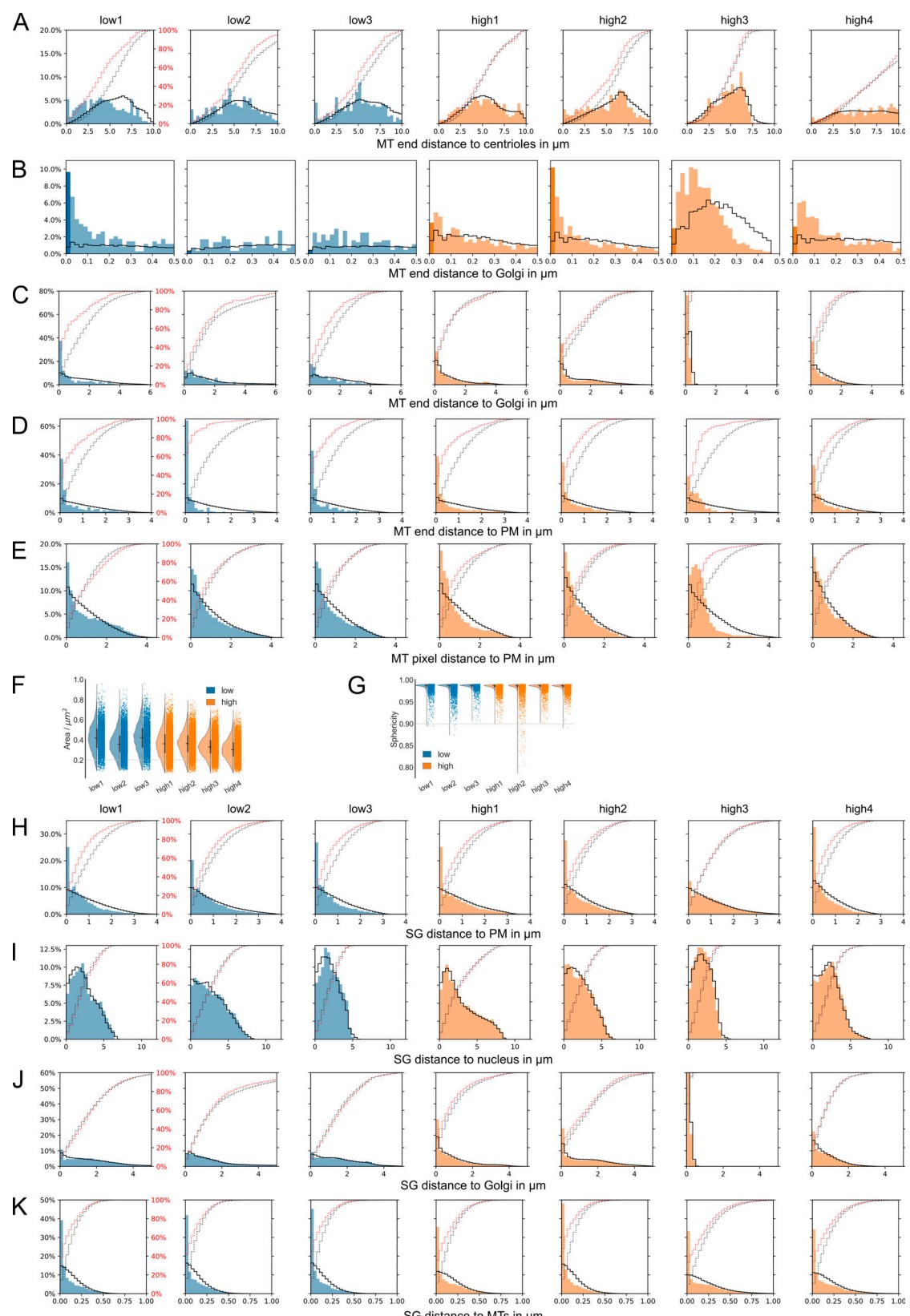

Figure S3. **Microtubule and SG analysis for all cells. (A)** Distance of microtubule ends to the nucleus. **(B)** Distance of microtubule ends to Golgi (bin 20 nm). **(C)** Distance of microtubule ends to Golgi (bin 200 nm). **(D)** Distance of microtubule ends to plasma membrane. **(E)** Distance of microtubule pixels to plasma membrane. **(F)** Surface areas of SGs. **(G)** Sphericity of SGs. **(H)** Distance of SGs to plasma membrane. **(I)** Distance of SGs to nucleus. **(J)** Distance of SGs to Golgi apparatus. **(K)** Distance of SGs to microtubules. Black lines in all distance distribution plots show the respective random distributions. Red dotted and black dotted lines represent actual and random cumulative distributions, respectively. MT, microtubule; PM, plasma membrane.

Video 1.   **Raw FIB volume.** Raw FIB-SEM volume corresponding to high-glucose cell 1. Walking through the volume in z-direction reveals ultrastructural details such as insulin SGs, mitochondria, ER, ribosomes, the Golgi apparatus, the nucleus, and microtubules. Scale bar, 500 nm.

Video 2.   **β cell microtubule and organelle segmentation.** Raw FIB-SEM and corresponding 3D segmentation of high-glucose cell 1. Segmented plasma membrane (gray transparent), insulin SGs (orange), mitochondria (blue), Golgi apparatus (green), nucleus (yellow), centrioles (purple), and microtubules (red). Removal of insulin SGs, mitochondria, and Golgi apparatus reveals the β cell microtubule network. Rendering was done with ORS Dragonfly.

Video 3.   **BetaSeg Viewer.** Demonstration of the functionality of the BetaSeg Viewer Fiji plugin. It allows loading of segmentation masks together with raw image stacks and navigation through the data. The user can visualize the different insulin SG categories and even obtain data of individual SGs. MT, microtubule; PM, plasma membrane.

Video 4.   **Microtubule network and centrosomal microtubules.** 3D rendering of one mouse β cell (low-glucose cell 3) with segmentation of centrosomal microtubules (red) in comparison to all microtubules (red) of one β cell together with nucleus (yellow) and centrioles/axoneme (purple). Removal of non-centrosomal microtubules reveals the centrosomal microtubule subset of the cell. Rendering was done with ORS Dragonfly.

Video 5.   **Microtubule network and Golgi-connected microtubules.** 3D rendering of one mouse β cell (high-glucose cell 1) with segmentation of Golgi-connected microtubules (red) in comparison to all microtubules (red) of one β cell together with Golgi apparatus (green) and nucleus (yellow). Removal of non-Golgi microtubules reveals the Golgi-connected microtubule subset of the cell. Rendering was done with ORS Dragonfly.

Video 6.   **Microtubule associated and not associated insulin SGs.** 3D rendering of one mouse β cell (high-glucose cell 2) with segmentation of microtubule-associated (orange) and –not associated insulin SGs (light gray) together with microtubules (red). Removal of not associated SGs reveals the subset of microtubule connected SGs. Rendering was done with ORS Dragonfly.

Video 7.   **3D animation of all segmented β cells.** Animation showing a raw FIB-SEM slice followed by the corresponding segmentation masks and a 3D rendering of the respective reconstructed β cell (high-glucose cell 1) with a transparent plasma membrane, insulin SGs (orange), mitochondria (blue), Golgi apparatus (green), centrioles (purple), nucleus (white), and microtubules (red). The individual organelles are highlighted followed by 3D renderings of all segmented cells. Rendering was done with blender. Videos can be viewed and downloaded via https://betaseg.github.io/.

