## [Peer Review File · The Journal of Cell Biology]

3D FIB-SEM reconstruction of microtubule-organelle interaction in whole primary mouse beta cells

Andreas Müller, Deborah Schmidt, C. Shan Xu, Song Pang, Joyson D'Costa, Susanne Kretschmar, Carla Münster, Thomas Kurth, Florian Jug, Martin Weigert, Harald Hess, and Michele Solimena

Corresponding Author(s): Andreas Müller, University Hospital and Faculty of Medicine Carl Gustav Carus, TU Dresden, Paul Langerhans Institute Dresden (PLID)

Review Timeline:	Submission Date:	2020-10-07
	Editorial Decision:	2020-11-02
	Revision Received:	2020-11-14

Monitoring Editor: Eva Nogales

Scientific Editor: Melina Casadio

Transaction Report:

DOI: <https://doi.org/10.1083/jcb.202010039>

November 2, 2020

RE: JCB Manuscript #202010039

Dr. Andreas Mueller
Paul Langerhans Institute Dresden (PLID) of the Helmholtz Center Munich at TU Dresden
Molecular Diabetology
Fetscherstrasse 74
Dresden 01307
Germany

Dear Dr. Mueller,

Thank you for submitting your manuscript entitled "Three-dimensional FIB-SEM reconstruction of microtubule-organelle interaction in whole primary mouse beta cells". Three expert referees evaluated the manuscript, and their comments are appended below. You will see that the reviewers praised the quality of the work and made suggestions to improve the data presentation, analysis, and discussion. We agree with the reviewers that the study is a technical tour-de-force that provides important new insight into the organization of pancreatic beta cells. Although the reviewers' comments should not require new experimentation, in the interest of clarity in particular, their points should be addressed rigorously and in full. We would be happy to publish your paper in JCB pending revisions necessary to meet our formatting guidelines (see details below) and to address the reviewers' comments. At resubmission, please include a point-by-point response to the reviews and please highlight the changes made to the manuscript text.

1) eTOC summary: A 40-word summary that describes the context and significance of the findings for a general readership should be included on the title page. The statement should be written in the present tense and refer to the work in the third person.

- Please include a summary statement on the title page of the resubmission. It should start with "First author name(s) et al..." to match our preferred style.

2) Tables should be provided as editable, individual files (e.g., Word, Excel).

3) JCB Reports are limited to 5 main and 3 supplementary figures. Could you please try to rearrange the data to meet this limit? Each figure can span up to one entire page, as long as all panels fit on the page.

4) Figure formatting: Scale bars must be present on all microscopy images, including inset magnifications. Would it be possible to please add scale bars to the reconstructions/renderings (2ACD, 3B, figure 4, S2)?

5) Statistical analysis: Error bars on graphic representations of numerical data must be clearly described in the figure legend. The number of independent data points (n) represented in a graph must be indicated in the legend. Statistical methods should be explained in full in the materials and

methods. For figures presenting pooled data the statistical measure should be defined in the figure legends.

6) Materials and methods: Should be comprehensive and not simply reference a previous publication for details on how an experiment was performed. Please provide full descriptions in the text for readers who may not have access to referenced manuscripts.

- For all cell lines, vectors, constructs/cDNAs, etc. - all genetic material: please include database / vendor ID (e.g., Addgene, ATCC, etc.) or if unavailable, please briefly describe their basic genetic features *even if described in other published work or gifted to you by other investigators*
- Microscope image acquisition: The following information must be provided about the acquisition and processing of images:
 - a. Make and model of microscope
 - b. Type, magnification, and numerical aperture of the objective lenses
 - c. Temperature
 - d. imaging medium
 - e. Fluorochromes
 - f. Camera make and model
 - g. Acquisition software
 - h. Any software used for image processing subsequent to data acquisition. Please include details and types of operations involved (e.g., type of deconvolution, 3D reconstitutions, surface or volume rendering, gamma adjustments, etc.).

7) References: There is no limit to the number of references cited in a manuscript. References should be cited parenthetically in the text by author and year of publication.

- Please abbreviate the names of journals according to PubMed.

8) A summary paragraph of all supplemental material should appear at the end of the Materials and methods section. Please include one brief sentence per item.

A. MANUSCRIPT ORGANIZATION AND FORMATTING:

Full guidelines are available on our Instructions for Authors page, <https://jcb.rupress.org/submission-guidelines#revised>. **Submission of a paper that does not conform to JCB guidelines will delay the acceptance of your manuscript.**

B. FINAL FILES:

-- High-resolution figure and video files: See our detailed guidelines for preparing your production-ready images, <https://jcb.rupress.org/fig-vid-guidelines>.

-- Cover images: If you have any striking images related to this story, we would be happy to consider them for inclusion on the journal cover. Submitted images may also be chosen for highlighting on the journal table of contents or JCB homepage carousel. Images should be uploaded

as TIFF or EPS files and must be at least 300 dpi resolution.

****It is JCB policy that if requested, original data images must be made available to the editors. Failure to provide original images upon request will result in unavoidable delays in publication. Please ensure that you have access to all original data images prior to final submission.****

****The license to publish form must be signed before your manuscript can be sent to production. A link to the electronic license to publish form will be sent to the corresponding author only. Please take a moment to check your funder requirements before choosing the appropriate license.****

Thank you for your attention to these final processing requirements. Please revise and format the manuscript and upload materials within 7-14 days. If complications arising from measures taken to prevent the spread of COVID-19 will prevent you from meeting this deadline (e.g. if you cannot retrieve necessary files from your laboratory, etc.), please let us know and we can work with you to determine a suitable revision period.

Thank you for this interesting contribution, we look forward to publishing your paper in Journal of Cell Biology.

Sincerely,

Eva Nogales, PhD
Monitoring Editor, Journal of Cell Biology

Melina Casadio, PhD
Senior Scientific Editor, Journal of Cell Biology

Reviewer #1 (Comments to the Authors (Required)):

The article by Müller et al. presents a 3D FIB-SEM reconstruction of pancreatic beta cells treated with low or high insulin concentrations prior to fixation. The application of FIB-SEM to these cells enabled the authors to characterize in detail the microtubule structure and its relation to insulin secretory granules and other organelles. The images are beautiful, the quantification is appropriate and is controlled for the overall geometry of the cells, and the supplementary videos are striking. The article is a technical tour-de-force and will serve as a model for future studies using FIB-SEM. Only 3 basal and 4 glucose-stimulated cells were quantified, and one of the glucose-stimulated cells had signs of stress, but the data nonetheless demonstrate new details about organelle structure and beta cell variability. In particular, the data address longstanding questions about how glucose-stimulated insulin secretion acts to cause insulin granule exocytosis, in part by remodeling the microtubule cytoskeleton. Contrary to the idea that most microtubules originate at the TGN in beta cells, the authors find that 80% of the microtubules in the cells are not anchored at either the Golgi or at centrosomes. Insulin caused a decrease in average microtubule length, an increase in the number of microtubules, and no change in microtubule density, suggesting possible effects on severing enzymes. The majority of the insulin granules were near the PM; after glucose stimulation there was an increase in the number of granules observed near the Golgi, consistent with formation of new granules at the TGN. The images also provide substantial quantitative description of other organelles. In my opinion, the article will be of broad interest and is appropriate for JCB.

I have only one minor comment which is that it would be helpful to plot the existing quantification of some of the data in an additional way. For the plots in Figs. 2E-H, Fig. 3D-F, and Suppl. Fig. 4, it would be helpful to see a trace for the cumulative data (i.e. the running integral of the shaded areas, and possibly of the random trace as well, along the x axis in each of the panels shown). This would make it easier to know at a glance, for example, what percentage of the microtubule ends are within X distance (e.g. 1 μm) of the PM. If superimposing such cumulative data quantifications on the existing plots makes them too confusing, then these could be included as a supplemental figure.

Reviewer #2 (Comments to the Authors (Required)):

This paper employs state of the art methods for sample preparation, electron microscopy at modest but adequate 3D resolution, and segmentation routines for extracting meaningful organelle positions from image data. These tools are used to characterize organelle position and microtubule network structure in beta cells from pancreatic islets. Image data in the paper and supplementary videos support the authors' contention that the cells are well preserved and therefore worthy of detailed study. The EM method (FIB-SEM) is beautifully employed, as demonstrated by Video 1, giving confidence that the several cells examined have been faithfully reconstructed at the resolution attainable with this technology (far better than "super-resolution" light microscopy, although somewhat less good than axial electron tomography). The effectiveness and accuracy of the segmentation methods used are defended by the quality of the images and movies shown, and reasonable statistical methods are used to compute the distributions of distances between cellular features of interest. The displays of image and graphic data are clear and convincing. The descriptions include several beta cells, frozen for study under two important conditions of stimulation by extracellular glucose. In short, this paper is a technological tour-de-force and deserves special commendation for being an excellent piece of structural biology.

The results obtained provide good evidence for the conclusions stated by the authors in both their abstract and their summary: centrosome or Golgi complex association explains only a small fraction of the microtubules in these fully differentiated cells (based on well-analyzed information about distances between the potential initiators and microtubule ends). The microtubule array is extensive, tortuous, and appears to place most of the polymers fairly near the plasma membrane. The authors recognize that the structure observed does not inform us about the genesis of microtubule array, but it is informative about how these mature cells can use microtubules for the task at hand: regulated granule secretion. The changes in microtubule arrangement with glucose stimulation are not extensive, but they are interesting. The fact that the secretory granules are continuously MT-associated and near the plasma membranes is informative about how these cells remain poised for efficient secretion, whether stimulated or not.

The segmentation of secretory granules and mitochondria appears to be excellent. The methods section mentions the software employed in each case, and the images in the paper are convincing that the methods were effective. As someone who has confronted this problem himself, I am much impressed, but I do wish that the paper said more about exactly how the segmentations worked. I checked out some of the relevant references, and I can imagine that some editing was required to produce the elegant images shown here. I note that the segmentation of the Golgi complex seemed to be much less successful than that of the structurally simpler organelles, and this is worthy of some comment. Both the figure representations in the paper and the videos show the

Golgi as far less-well defined than in the segmentations obtained from electron axial tomography (e.g., Marsh et al., 98:2399-2406(2001)). A brief comment about the difficulty of extracting membranes from a tortuous organelle, rather than something more spherical might help the reader to understand the limitations of the current study, impressive as it is.

One reaction to this paper is that the authors did a great deal of work to show that neither the secretory granules nor the microtubules change their arrangements very much in response to a stimulus to secrete. In a narrow-minded way, this is disappointing. On the other hand, the technologies employed here seem to this reader to be perfect for the task undertaken, and the results are very clear: regulation of secretion rate is not at the level of a major change in the arrangement of the cytoskeleton. This is very good to know, and I don't see another way of learning this answer. In this reader's opinion, this paper should be published, with only minor modifications, because of its authoritative use of all relevant methods, the beauty of the resulting structures, and the solid facts it presents, even if they are not spectacular.

Specific Comments

Fig 1A Put axis labels onto the block of tissue to explain the XY and XZ labels in the tomographic slices.

1B. The field of view is the same for the first four images, then changes in both area and magnification. This is mentioned for centrioles but not for the MT image.

Fig 2A The purple centrioles don't stand out very well against the MTs. Perhaps make them orange or yellow?

Fig 4 legend. First sentence ends with "and microtubules (Orange)", but as previously stated and in the images, the MTs appear red.

Table 1 and text P4. MT lengths are reported with four significant figures but length totals to 5. Drop the decimal points figures.

The methods section includes the statement, "When plotting the distance of microtubule ends to the Golgi, the second histogram line in black accounts for the distances of all pixels in the cell to the Golgi, excluding any pixels belonging to the nucleus or to the Golgi itself. If microtubule ends would be randomly located in the cell and if their positions would not correlate with the location of other organelles, their distribution should roughly match this plot."

I understand and agree with this, but if the point were made in the text when these graphs with the "random" lines are first shown, the significance of the high bars at low distances would be more evident to the reader.

Reviewer #3 (Comments to the Authors (Required)):

This paper is a technical tour de force. Müller et al. have used FIB-SEM to image and reconstruct the entire microtubule cytoskeleton of 6 pancreatic islet beta cells, plus the Golgi apparatus, centrioles, mitochondria, secretory granules, and the nucleus. The resulting 3D renders of cells are extraordinary (see Fig. 1C). Importantly, the authors produce 3D renders for beta cells plus/minus glucose, so they can fully reconstruct the response to a key stimulus. The authors use these volumes to make several remarkable observations about organization of the cytoskeleton and microtubule-organelle interactions. The paper should absolutely be published with revisions.

Comments below:

-- Polymer homeostasis

The authors show that the total amount of microtubule polymer is constant with glucose stimulation. High-glucose cells have approximately 3X the number of microtubules as low-glucose cells, but they are 1/3 as long. This is a very cool observation of homeostasis of monomer/polymer levels, but the time-scale and mechanism of this homeostasis could be discussed in more detail. The glucose treatment is 1 hour, but Zhu et al. Dev Cell 2015 imaged microtubules within the first minutes of glucose treatment and saw nucleation from the Golgi. Are you seeing the end-point of what Zhu et al. saw as initial responses? In the Summary, you contrast your work strongly with Zhu et al., and cast doubt on their conclusions. Are the discrepancies explainable by differences in time-scale?

On a related note, I strongly dislike the term "microtubule density". What you mean is "total polymer density". To me, microtubule density = microtubule number / volume. So it's not true that microtubule density stays the same, because microtubule number changes with glucose. It's total polymer that remains constant. Please use "polymer density".

-- Golgi connection

The microtubules are manually traced. Was any effort made to ensure reproducibility of this tracing between researchers? Or are we trusting one person's eye? I've seen papers from Jan Funke's group about semi-automated microtubule-specific detection methods (e.g., on the arXiv), but I'm not a specialist so I don't know if they are readily applicable. I recommend more detail in this section of the Methods, including a description of whatever safeguards are put in place to prevent bias.

On a related note, how accurate is your positioning of the microtubule ends? I can imagine that the SNR is reduced at microtubule ends, for example if the protofilaments become tapered. Or maybe the ends terminate abruptly, which would be something worth mentioning.

The accuracy of end-positioning affects how we understand the data on connections to the Golgi. A microtubule is considered "Golgi-connected" if the end is 20 nm from the Golgi. That's less than 3 rows of tubulin dimers. It's easy for me to imagine errors in end-positioning that occur on this scale.

Also: why 20 nm? Is that number based on the structure of a tethering protein or something? Could tethering proteins be non-globular or extended? Could a microtubule end be 40 nm away from the Golgi and still "connected"? This metric feels arbitrary.

Maybe the answer is to just look at the data. I can't really make out the bin size in Fig. 2E and 2F, but the smallest bin looks to be about 100 nm. In the low-glucose case of Fig. 2F ("low 1"), the smallest bin is nearly 40% of the total. But in Table 1, the highest percentage of "Microtubule Golgi-connected" is 9.6%. If you ask me, Fig. 2F looks like you have 35% of your population in a "connected" state and the remaining 65% spread over a distribution of distances. Could a fitting procedure be used to identify whether the data appears to contain two populations?

-- Nucleation sites

In any case, the percentage of microtubules that are NOT connected to their nucleation site was very surprising. By the authors' estimates, 9/10 microtubules are NOT connected to the places

where the g-TuRCs are thought to be located (PCM and Golgi membranes). The authors should certainly discuss this observation further, because it has major implications for nucleation and minus-end dynamics. Does the g-TuRC break free from the Golgi after nucleation? Is the microtubule severed at the minus end and then capped by CAMSAPs? Or are the active nucleation factors NOT where we think they are?

In summary, this paper made me very excited for the future of FIB-SEM and for our understanding of the way different cell types control their cytoskeletons. Bravo.

We thank all the reviewers for their valuable input and are especially happy that all of them share our excitement about our work. We are grateful for the advice and detailed comments provided by the reviewers that improved several aspects of our manuscript and helped to place our findings into a broader context.

In the revised manuscript we have now

- Included cumulative distributions for all data,
- included additional information on our segmentation methods, and
- expanded the discussion on microtubule nucleation sites, according to the reviewer's suggestions (highlighted in red in the manuscript).
- Additionally we joined Supplementary Figure 1 and 3 as per the journal guidelines and added scale-cubes to all 3D renderings.

We hope that this addresses all outstanding issues.

Please find in the following our point-by-point response to the reviewers comments (highlighted in blue)

Reviewer #1 (Comments to the Authors (Required)):

The article by Müller et al. presents a 3D FIB-SEM reconstruction of pancreatic beta cells treated with low or high insulin concentrations prior to fixation. The application of FIB-SEM to these cells enabled the authors to characterize in detail the microtubule structure and its relation to insulin secretory granules and other organelles. The images are beautiful, the quantification is appropriate and is controlled for the overall geometry of the cells, and the supplementary videos are striking. The article is a technical tour-de-force and will serve as a model for future studies using FIB-SEM. Only 3 basal and 4 glucose-stimulated cells were quantified, and one of the glucose-stimulated cells had signs of stress, but the data nonetheless demonstrate new details about organelle structure and beta cell variability. In particular, the data address longstanding questions about how glucose-stimulated insulin secretion acts to cause insulin granule exocytosis, in part by remodeling the microtubule cytoskeleton. Contrary to the idea that most microtubules originate at the TGN in beta cells, the authors find that 80% of the microtubules in the cells are not anchored at either the Golgi or at centrosomes. Insulin caused a decrease in average microtubule length, an increase in the number of microtubules, and no change in microtubule density, suggesting possible effects on severing enzymes. The majority of the insulin granules were near the PM; after glucose stimulation there was an increase in the number of granules observed near the Golgi, consistent with formation of new granules at the TGN. The images also provide substantial quantitative description of other organelles. In my opinion, the article will be of broad interest and is appropriate for JCB.

I have only one minor comment which is that it would be helpful to plot the existing quantification of some of the data in an additional way. For the plots in Figs. 2E-H, Fig. 3D-F, and Suppl. Fig. 4, it would be helpful to see a trace for the cumulative data (i.e. the running integral of the shaded areas, and possibly of the random trace as well, along the x axis in each of the panels shown). This would make it easier to know at a glance, for example, what percentage of the microtubule ends are within X distance (e.g. 1 μ m) of the PM. If superimposing such cumulative data quantifications on the existing plots makes them too confusing, then these could be included as a supplemental figure.

We thank the reviewer for this comment and agree that including cumulative distributions would be helpful. We therefore added the cumulative distributions to all the plots of both the measured and random values which especially helps to address differences regarding actual and random distributions. We added lines for the cumulative data to most of the plots in the new Supplementary Figure 3 as well as all of the plots in Figure 2, 3 and 4.

Reviewer #2 (Comments to the Authors (Required)):

This paper employs state of the art methods for sample preparation, electron microscopy at modest but adequate 3D resolution, and segmentation routines for extracting meaningful organelle positions from image data. These tools are used to characterize organelle position and microtubule network structure in beta cells from pancreatic islets. Image data in the paper and supplementary videos support the authors' contention that the cells are well preserved and therefore worthy of detailed study. The EM method (FIB-SEM) is beautifully employed, as demonstrated by Video 1, giving confidence that the several cells examined have been faithfully reconstructed at the resolution attainable with this technology (far better than "super-resolution" light microscopy, although somewhat less good than axial electron tomography). The effectiveness and accuracy of the segmentation methods used are defended by the quality of the images and movies shown, and reasonable statistical methods are used to compute the distributions of distances between cellular features of interest. The displays of image and graphic data are clear and convincing. The descriptions include several beta cells, frozen for study under two important conditions of stimulation by extracellular glucose. In short, this paper is a technological tour-de-force and deserves special commendation for being an excellent piece of structural biology.

The results obtained provide good evidence for the conclusions stated by the authors in both their abstract and their summary: centrosome or Golgi complex association explains only a small fraction of the microtubules in these fully differentiated cells (based on well-analyzed information about distances between the potential initiators and microtubule ends). The microtubule array is extensive, tortuous, and appears to place most of the polymers fairly near the plasma membrane. The authors recognize that the structure observed does not inform us about the genesis of microtubule array, but it is informative about how these mature cells can use microtubules for the task at hand: regulated granule secretion. The changes in microtubule arrangement with glucose stimulation are not extensive, but they are interesting. The fact that the secretory granules are continuously MT-associated and near the plasma membranes is informative about how these cells remain poised for efficient secretion, whether stimulated or not.

The segmentation of secretory granules and mitochondria appears to be excellent. The methods section mentions the software employed in each case, and the images in the paper are convincing that the methods were effective. As someone who has confronted this problem himself, I am much impressed, but I do wish that the paper said more about exactly how the segmentations worked. I checked out some of the relevant references, and I can imagine that some editing was required to produce the elegant images shown here. I note that the segmentation of the Golgi complex seemed to be much less successful than that of the structurally simpler organelles, and this is worthy of some comment. Both the figure representations in the paper and the videos show the Golgi as far less-well defined than in the segmentations obtained from electron axial tomography (e.g., Marsh et al., 98:2399-2406(2001)). A brief comment about the difficulty of extracting membranes from a tortuous organelle, rather than something more spherical might help the reader to understand the limitations of the current study, impressive as it is.

This is a very interesting comment. Brad Marsh's Golgi reconstructions are indeed truly amazing and unprecedented to date. We have now included the paper in the references and discuss it in the results together with our own Golgi segmentation. Since our goal was to reconstruct the whole Golgi to define possible sites of microtubule nucleation we did not segment Golgi vesicles, but rather focused on Golgi cisternae. Also, in Marsh et al., 2001 not the whole Golgi, but a part of it, is reconstructed and cisternae are cut, revealing beautifully the lumen of the Golgi complex. Our segmentations are closed, which makes the Golgi cisternae appear not as text-book like. Our results are comparable with the Golgi reconstructions in Noske et al., 2007 (also with Brad Marsh) showing large perinuclear complexes. Compared to these data our segmentations are more precise since we did not introduce interpolations between axial sections.

One reaction to this paper is that the authors did a great deal of work to show that neither the secretory granules nor the microtubules change their arrangements very much in response to a stimulus to secrete. In a narrow-minded way, this is disappointing. On the other hand, the technologies employed here seem to this reader to be perfect for the task undertaken, and the results are very clear: regulation of secretion rate is not at the level of a major change in the arrangement of the cytoskeleton. This is very good to know, and I don't see another way of learning this answer. In this reader's opinion, this paper should be published, with only minor modifications, because of its authoritative use of all relevant methods, the beauty of the resulting structures, and the solid facts it presents, even if they are not spectacular.

Specific Comments

Fig 1A Put axis labels onto the block of tissue to explain the XY and XZ labels in the tomographic slices.

Thanks for the suggestion. We added axis labels to the tissue block.

1B. The field of view is the same for the first four images, then changes in both area and magnification. This is mentioned for centrioles but not for the MT image.

We added a sentence clarifying this to the Figure legend.

Fig 2A The purple centrioles don't stand out very well against the MTs. Perhaps make them orange or yellow?

Thanks for the suggestion. We modified the centriole color in Fig 2A, so they should now be easier to spot.

Fig 4 legend. First sentence ends with "and microtubules (Orange)", but as previously stated and in the images, the MTs appear red.

Thanks for catching this mistake. We corrected this.

Table 1 and text P4. MT lengths are reported with four significant figures but length totals to 5. Drop the decimal points figures.

We changed the decimal points in the text according to the values presented in T1.

The methods section includes the statement, "When plotting the distance of microtubule ends to the Golgi, the second histogram line in black accounts for the distances of all pixels in the cell to the Golgi, excluding any pixels belonging to the nucleus or to the Golgi itself. If microtubule ends would be randomly located in the cell and if their positions would not correlate with the location of other organelles, their distribution should roughly match this plot."

I understand and agree with this, but if the point were made in the text when these graphs with the "random" lines are first shown, the significance of the high bars at low distances would be more evident to the reader.

We agree with the reviewer and added additional information to the results that was before in the methods section.

Reviewer #3 (Comments to the Authors (Required)):

This paper is a technical tour de force. Müller et al. have used FIB-SEM to image and reconstruct the entire microtubule cytoskeleton of 6 pancreatic islet beta cells, plus the Golgi apparatus, centrioles, mitochondria, secretory granules, and the nucleus. The resulting 3D renders of cells are extraordinary (see Fig. 1C). Importantly, the authors produce 3D renders for beta cells plus/minus glucose, so they can fully reconstruct the response to a key stimulus. The authors use these volumes to make several remarkable observations about organization of the cytoskeleton and microtubule-organelle interactions. The paper should absolutely be published with revisions. Comments below:

-- Polymer homeostasis

The authors show that the total amount of microtubule polymer is constant with glucose stimulation. High-glucose cells have approximately 3X the number of microtubules as low-glucose cells, but they are 1/3 as long. This is a very cool observation of homeostasis of monomer/polymer levels, but the time-scale and mechanism of this homeostasis could be discussed in more detail. The glucose treatment is 1 hour, but Zhu et al. Dev Cell 2015 imaged microtubules within the first minutes of glucose treatment and saw nucleation from the Golgi. Are you seeing the end-point of what Zhu et al. saw as initial responses? In the Summary, you contrast your work strongly with Zhu et al., and cast doubt on their conclusions. Are the discrepancies explainable by differences in time-scale?

We thank the reviewer for these valuable comments. Zhu et al. and our work differ in their experimentation and thus are not describing the exact same situation: We show the response of the beta cell upon exposure to high glucose for 1 hr, whereas Zhu et al. show what happens after all microtubules have been first pharmacologically depolymerized by Nocodazole treatment for 12 hrs and then allowed to recover. Thus, we believe that our observations are not taken at an end-point, but reflect a more physiological condition. Also, Zhu et al were not able to determine whether microtubules stay at their respective nucleation sites (which could be possible to observe with vastly longer high resolution imaging). Regarding the investigation of tubulin density our data are comparable since we use similar protocols for islet stimulation.

We have included a remark on the nocodazole treatment in Zhu et al. in the discussion section and discuss possible mechanisms leading to shorter, disconnected microtubules later in the text..

On a related note, I strongly dislike the term "microtubule density". What you mean is "total polymer density". To me, microtubule density = microtubule number / volume. So it's not true that microtubule density stays the same, because microtubule number changes with glucose. It's total polymer that remains constant. Please use "polymer density".

Thanks for the comment. We agree that "microtubule density" is for our case not the precise term and changed it to "tubulin polymer density" in the text.

-- Golgi connection

The microtubules are manually traced. Was any effort made to ensure reproducibility of this tracing between researchers? Or are we trusting one person's eye? I've seen papers from Jan Funke's group about semi-automated microtubule-specific detection methods (e.g., on the arXiv), but I'm not a specialist so I don't know if they are readily applicable. I recommend more detail in this section of the Methods, including a description of whatever safeguards are put in place to prevent bias.

Microtubules were traced by two authors JVD'C and AM, with AM curating the results and applying final corrections for all cells. We now included a paragraph to the method section describing this. We have tried different approaches for automatic MT segmentation including deep learning. However, this did not result in satisfactory segmentations most likely due to the low contrast of MT in our images. We furthermore tried to use the microtubule tracing in Amira (see Weber et al., 2013), which is used for tracing microtubules in electron tomography volumes of mitotic spindles. Interestingly, the program managed to detect parts of the microtubules. However, since our microtubules had a higher curvature than spindle microtubules, Amira never traced whole microtubules, which would have made a lot of manual curation necessary. Since Amira is also not ideal for the handling of large datasets, we decided to continue manual tracing with Knossos because of its data cubing abilities.

We are aware of Jan Funke's work and find the latest preprint very interesting, although not readily applicable to our datasets. We think that deep learning will help in the future to segment microtubules in a variety of samples, as it helped us for segmentation of other organelles. However, in our high-pressure frozen samples the contrast was lower than in chemically fixed brain tissue for connectomics, as used in the Funke lab latest work. This for now probably hinders deep learning tracing and could be overcome by the generation of more ground-truth or further increasing sample contrast.

On a related note, how accurate is your positioning of the microtubule ends? I can imagine that the SNR is reduced at microtubule ends, for example if the protofilaments become tapered. Or maybe the ends terminate abruptly, which would be something worth mentioning.

In most cases we found that microtubules ended abruptly, which we would expect for (stabilized) minus-ends. Only in a few cases we found flared microtubule ends. Therefore, we feel confident that in the majority of cases microtubule ends could be positioned precisely. We added a sentence to this to the method section and added example images of microtubule ends to Supplementary Figure 1 C.

The accuracy of end-positioning affects how we understand the data on connections to the Golgi. A microtubule is considered "Golgi-connected" if the end is 20 nm from the Golgi. That's less than 3 rows of tubulin dimers. It's easy for me to imagine errors in end-positioning that occur on this scale.

Also: why 20 nm? Is that number based on the structure of a tethering protein or something? Could tethering proteins be non-globular or extended? Could a microtubule end be 40 nm away from the Golgi and still "connected"? This metric feels arbitrary.

According to Chabrin-Brion et al., 2001, Efimov et al., 2007 and Sanders and Kaverina, 2015 Golgi microtubules originate directly from Golgi membranes. We therefore categorized microtubules as Golgi-connected, when a microtubule end would be in contact or next to Golgi membranes and used the 20 nm threshold to correct for inaccuracies in the segmentation. We now include those citations and indicate some of the possible mechanisms for microtubule polymerization at Golgi membranes (g-TuRC stabilization by CLASP, binding mediated by AKAP). Furthermore, g-TuRC as well proteins that decorate microtubules can be most likely not be discriminated from microtubules at our resolution. They are therefore included within the microtubule segmentations.

Maybe the answer is to just look at the data. I can't really make out the bin size in Fig. 2E and 2F, but the smallest bin looks to be about 100 nm. In the low-glucose case of Fig. 2F ("low 1"), the smallest bin is nearly 40% of the total. But in Table 1, the highest percentage of "Microtubule Golgi-connected" is 9.6%. If you ask me, Fig. 2F looks like you have 35% of your population in a "connected" state and the remaining 65% spread over a distribution of distances. Could a fitting procedure be used to identify whether the data appears to contain two populations?

This is a very good point. There is indeed further information in the data that could not be seen with the current bin size. We thus now modified Supplementary Figure 3 B showing the distance distribution of MT ends vs Golgi with a bin size of 20nm. Interestingly, there was a high heterogeneity between cells. Whereas low glucose cell 1, high glucose cell 2 and 4 showed high densities of microtubule ends in the area close to the Golgi membranes,

for low glucose cell 2 and 3 and high glucose cell 1 this was not the case. We discuss this further in the text with our interpretation being that the high density indicates that these microtubules have just recently detached from the Golgi membranes and are still in close vicinity.

-- Nucleation sites

In any case, the percentage of microtubules that are NOT connected to their nucleation site was very surprising. By the authors' estimates, 9/10 microtubules are NOT connected to the places where the g-TuRCs are thought to be located (PCM and Golgi membranes). The authors should certainly discuss this observation further, because it has major implications for nucleation and minus-end dynamics. Does the g-TuRC break free from the Golgi after nucleation? Is the microtubule severed at the minus end and then capped by CAMSAPs? Or are the active nucleation factors NOT where we think they are?

Thanks for these comments which we find very interesting and exciting! We think that our findings have implications for two major aspects in the biology of beta cells: Insulin trafficking in mature cells and beta cell development and maturation. It would be very interesting to investigate these topics further and to look into the role of minus end stabilizers (CAMSAP), anchors (Ninein) and severing enzymes in insulin secretions as well as beta cell/pancreas development. Is the activity of some of these proteins important for insulin secretion? Is major microtubule remodelling happening during beta cell development? Are there possible other sites of microtubule generation besides Golgi and centrioles or are microtubules generated only there followed by severing? We thank the reviewer for pointing this out and added a paragraph to the discussion.

In summary, this paper made me very excited for the future of FIB-SEM and for our understanding of the way different cell types control their cytoskeletons. Bravo.

Thank you!